# PREFERENCE-DRIVEN SPATIAL-TEMPORAL COUNTING PROCESS MODELS

## ABSTRACT

Traditional spatial-temporal models often overlook the complex decision-making processes and social factors that shape spatial-temporal event data generated by humans. This paper introduces a novel framework that integrates choice theory with social intelligence to model and analyze counting processes, such as crime occurrences or bike-sharing activity, where the observed discrete events result from individual decisions influenced by social dynamics. Our approach aims to uncover latent human preference patterns, represented by utility functions, to capture the diverse decision-making factors within a population that result in the observed event counts. These latent factors help explain how choices—such as where and when to commit a crime—are shaped by personal preferences, environmental conditions, and social influences. By modeling the aggregate outcomes of these individual choices, we can better understand and predict patterns in counting processes. The proposed model adopts a preference-driven approach to counting data, providing interpretable insights at a detailed level. It also enables in-depth analysis of how external interventions, like law enforcement actions or policy changes, influence individual decisions and how these effects spread through the system. Empirical evaluation of crime and bike-sharing datasets demonstrates our model's ability to offer clear insights and achieve high predictive accuracy.

## 1 INTRODUCTION

Many real-world counting processes, such as criminal or bike-sharing activities, are generated by human behavior and result from a series of decision-making processes driven by individual preferences (Zhao & Tang, 2017; He et al., 2021). These spatial-temporal discrete events are not merely the outcome of spatial-temporal interactions but are significantly influenced by human factors, including diverse preferences and complex social dynamics. Traditional spatial-temporal models, such as those using Gaussian processes in the Log-Cox Gaussian Process model (Møller et al., 1998; Diggle et al., 2013) or incorporating triggering kernel functions in spatial-temporal point processes (Reinhart, 2018), focus primarily on capturing spatial-temporal dependencies. However, these models often fall short of addressing the underlying human decision-making processes and social influences that shape these events.

To address this gap, we propose an alternative modeling perspective that aims to uncover the latent utility functions driving the mechanisms behind human-generated discrete events. By incorporating these latent preferences and decision-making factors, our approach seeks to provide a more comprehensive and insightful framework for analyzing and predicting counting processes.

Our approach extends traditional models by integrating choice theory (Levin & Milgrom, 2004; Bentz & Merunka, 2000) and social intelligence into a framework that automatically discovers and explains patterns in counting processes. Specifically, we focus on modeling how individuals make choices—such as where and when to commit a crime—based on underlying factors that may include social influences, environmental cues, and personal preferences. Even though our data may be counts, such as the number of crimes in each location per hour, these counts reflect the aggregate outcomes of individual choices influenced by various factors.

At the core of our model is random utility function modeling (Azari et al., 2012; Aouad & Désir, 2022), which enables us to understand and quantify the preferences driving individual decisions. These preferences are influenced by a range of factors, including social norms, environmental cues,

and personal tastes. Our approach leverages a preference learning framework that utilizes a mixture-of-expert models (Jacobs et al., 1991; Shazeer et al., 2017), each representing a distinct preference pattern. This allows us to capture the diverse ways in which individuals decide where and when to generate an event.

Each preference function in our model is inspired by a two-stage decision-making process. In the first stage, we develop a ranking function to evaluate and prioritize all possible time-location pairs. This ranking function incorporates considerations of social norms and mutual influences—key aspects of social intelligence that reflect how people's choices are shaped by their interactions and the behavior of others. The ranking function is followed by a soft ranking-based sparse selection method, which uses a learned sparse gating function to adaptively select a subset of options (Peters et al., 2019; Correia et al., 2019). This approach allows the model to either spread out or focus on specific time-location pairs based on the spatial-temporal dynamics of the data.

In the second stage, we compare the selected options within the chosen subset using an additional set of learnable utility parameters. This step introduces further flexibility and detail into the decision-making process, enabling us to refine the final choice based on the previously ranked options.

Overall, our model captures the intricate patterns in spatial-temporal counting processes, revealing how human choices, driven by varying preferences and social influences, shape the distribution of events over time and space. The use of sparse selection and ranking functions, combined with a mixture-of-expert models, enhances the interpretability and adaptability of our approach, providing a detailed understanding of how different factors contribute to the decision-making process.

Our contributions are threefold. First, we present a novel human preference-driven model for spatial-temporal counting processes, capturing how individual choices influence the distribution of events over time and space. Second, our preference model is informed by social intelligence, incorporating flexible, adaptive techniques and offering interpretability through integrating social norms and mutual influences. Finally, we demonstrate the effectiveness of our model through empirical evaluation of real-world crime and bike-sharing datasets, validating its ability to uncover insightful patterns and improve predictive performance.

## 2 RELATED WORK

**Spatial-Temporal Modeling**   Spatial-temporal modeling involves analyzing and predicting the evolution of state or value across both spatial and temporal dimensions. Recent advances in deep learning methods enhance the capability of spatial-temporal modeling Luo et al. (2020); Zhang et al. (2021). Recurrent Neural Networks (RNNs) (Yu et al., 2017b; Wang et al., 2017) and Temporal Convolutional Networks (TCMs) Wu et al. (2019) demonstrating their effectiveness in handling temporal dynamics. To capture spatial dependencies, onvolutional neural networks (CNNs) (Li et al., 2022) and graph neural networks (GNNs) (Yu et al., 2017a) have been adopted to model spatial dependencies.

Spatial-Temporal Point Processes models also offer a robust methodology for handling the generative process of discrete events in continuous time and space through intensity functions, eliminating the necessity to partition space and time into cells (Moller & Waagepetersen, 2003; Diggle, 2006; Reinhart, 2018). The occurrence intensity of events is a function of space, time, and history, and explicitly characterizes how the events are allocated over time and space. The Log-Gaussian Cox process (LGCP), where the log intensity function is a random realization drawn from a Gaussian process (Møller et al., 1998; Diggle et al., 2013), although flexible, requires a pre-specified mean and covariance function to incorporate an accurate prior belief on the spacetime interleaved correlation. (Miller et al., 2014) utilized non-negative matrix factorization for mining low-rank spatial decompositions. Recent methods have improved the efficiency of fitting intensities through the use of neural networks. (Yuan et al., 2023) leveragsd diffusion models to learn complex spatio-temporal joint distributions. (Chen et al., 2020) applies Neural ODEs as the backbone, which parameterized the temporal intensity with Neural Jump SDEs and the spatial PDF with continuous-time normalizing flows. *However, the inherent incapacity to capture underlying patterns when constructing the intensity function results in a lack of interpretability.*

**Choice Model**   Choice models are used to explain or predict the choice behaviour of an individual or segment among a set of alternatives. The prevalent choice models include multinomial logit

model (MNL) (McFadden, 1972), Markov chain choice model (MCCM) (Blanchet et al., 2016), non-parametric choice model (NP) (Farias et al., 2009), mixture choice model (McFadden & Train, 2000), etc. The work of (Bentz & Merunka, 2000) add model flexibility by using neural network to capture the nonlinear effects of features on the latent utility. Recent studies such as (Wang et al., 2020; Han et al., 2020; Sifringer et al., 2020; Gabel & Timoshenko, 2022; Wang et al., 2023a; Arkoudi et al., 2023; Ko & Li, 2023; Wang et al., 2023b) have achieved high predictive performance by employing various deep neural network architectures to model complexs feature-to-utilty mappings. *We utilize the choice model as the building block of our proposed model.*

**Mixture-of-Experts** The concept of Mixture-of-Expert (MoE) was initially introduced by (Jacobs et al., 1991), which is based on a simple yet powerful idea: different parts of a model, known as experts, specialize in different tasks or aspects of the data. This approach achieved large-scale success when Shazeer et al. (2017) refined the gating mechanism which enhanced the capacity of MoE remarkably. Subsequent developments in the field have included novel advancements in gating design (Fedus et al., 2022; Lepikhin et al., 2020), optimization algorithms (Zoph et al., 2022), and distributed training frameworks (Rajbhandari et al., 2022). Recently, several studies adopt MoE for spatial-temporal prediction. For instance, Rahman et al. (2020) introduces a spatial-temporal MoE network with CNNs and RNNs as experts, applied to handle the source heterogeneity posed by multi-city ride-hailing demand prediction. Liu et al. (2023) constructs embeddings encoded with spatial-temporal knowledge and adopts MLPs as experts for prediction. Jiang et al. (2024) employs multiple tailored adaptive graph learners as experts to capture traffic congestion spatial-temporal patterns from various aspects and further introduces specialized experts to identify stable trends and periodic patterns from traffic data. *We aim to combine mixture-of-expert with choice model to capture more preference-guided patterns to explain event occurrence.*

## 3 BACKGROUND

Consider a spatial-temporal dataset where each event is captured as a tuple $(t_i, s_i)$, where $t_i$ represents the time of occurrence of the $i$-th event, and $s_i = (x_i, y_i)$ represents the spatial location of the event in a two-dimensional space. We first review two traditional modeling ideas as follows.

### 3.1 MODELING SPATIAL-TEMPORAL DATA WITH INTENSITY FUNCTION

Spatial-temporal point processes with a history-dependent intensity function are a rigorous and conventional framework for modeling such data. The key concept is the conditional intensity function, denoted as $\lambda(t, s \mid \mathcal{H}_t)$, which defines the rate at which events are expected to occur, given the history of past events:

$$\lambda(t, s \mid \mathcal{H}_t) = \lim_{\Delta t, \Delta s \to 0} \frac{\mathbb{E}\left[N((t, t + \Delta t] \times (s, s + \Delta s]) \mid \mathcal{H}_t\right]}{\Delta t \cdot \Delta s} \tag{1}$$

where $N((t, t+\Delta t] \times (s, s+\Delta s])$ is the number of events occurring in the infinitesimal time interval $(t, t + \Delta t]$ and spatial region $(s, s + \Delta s]$, and $\mathcal{H}_t$ denotes the history of the process up to time $t$, including all previous event times and locations.

The probability of observing $y$ events in cell $(t_i, s_i)$ is

$$P(N(t_i, s_i) = y) = \frac{(\lambda(t_i, s_i \mid \mathcal{H}_{t_i}))^y e^{-\lambda(t_i, s_i \mid \mathcal{H}_{t_i})}}{y!}$$

Given a set of observed events $\{(t_i, s_i)\}_{i=1}^N$, the log-likelihood of observing this data under the model defined by the conditional intensity function is:

$$\mathcal{L}\left(\{N(t_i, s_i) = y_i\}_{i=1}^N\right) = \sum_{i=1}^N y_i \log(\lambda(t_i, s_i \mid \mathcal{H}_{t_i})) - \lambda(t_i, s_i \mid \mathcal{H}_{t_i}) - \log(y_i!) \tag{2}$$

### 3.2 DISCRETE CHOICE MODEL: MODELING HUMAN'S CHOICE

We introduce the background regarding the choice model, which is an important building block of our model. The discrete choice model is a fundamental tool in economics and social sciences for

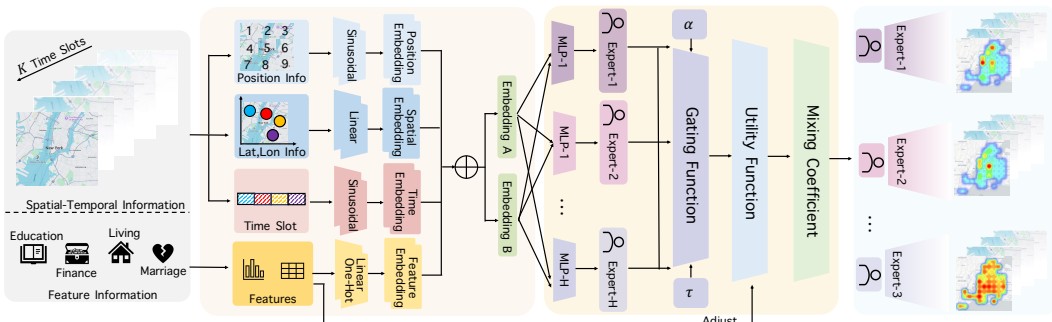

Figure 1: Model Framework.

modeling human decision-making processes. Specifically, the classic Plackett-Luce choice model (Plackett, 1975; Luce, 1959) uses a utility-based approach to model the choice probability. Each alternative $m \in [M]$ is associated with a utility $U_m$, and the probability of selecting option $m$ is given by the softmax function of the utilities:

$$\frac{\exp(U_m)}{\sum_{m'=1}^{M} \exp(U_{m'})}, \quad m \in [M], \tag{3}$$

The utility $U$ can depend on a variety of alternative-specific features, as well as individual-specific features.

When the number of alternatives $M$ is large, each individual may only consider a subset of the alternatives (Howard, 1969). To model this, a gating mechanism can be introduced to select a subset of alternatives before computing the choice probability. The gating mechanism can be implemented using a sparse vector $g \in \mathbb{R}^M$ that selects a subset of alternatives based on the input features. The choice probability can then be computed as

$$\frac{g_m \exp(U_m)}{\sum_{m'=1}^{M} g_{m'} \exp(U_{m'})}, \quad m \in [M]. \tag{4}$$

## 4    OUR MODEL: PREFERENCE-GUIDED SPATIAL-TEMPORAL CHOICE MODEL

To model spatial-temporal data driven by heterogeneous human preferences, we begin by partitioning both space and time into discrete spatial grids and time intervals, with each interval and grid represented by a central point, denoted as $(t_m, s_m)$. For instance, consider the decision-making process of a criminal selecting when and where to commit a crime. To capture this, we divide a day into time intervals (e.g., hours) and a city into spatial grids (e.g., blocks or neighborhoods). Suppose there are $M$ possible choices, $\{(t_m, s_m)\}_{m \in [M]}$. Each event involves the choice of a specific time and location. Our objective is to learn the underlying model that guide the choice in these events. To capture the probability of selecting a specific time-location pair for an event, we consider a gated latent class choice model of the form

$$\sum_{h=1}^{H} \pi^h \frac{g_m^h \exp(U_m^h)}{\sum_{m'=1}^{M} g_{m'}^h \exp(U_{m'}^h)}. \tag{5}$$

Here, the population is divided into $H$ latent classes, each with a different preference pattern, capturing the impact of temporal-spatial features on the choice. In this model, $\pi^h$ is the proportion of individuals in the $h$-th class, and $\sum_{h=1}^{H} \pi^h = 1$. For each class, $U^h \in \mathbb{R}^M$ is the utility vector for the $M$ time-location pairs, which can be either learnable parameters or a function of the learned embeddings. The gate vector $g^h \in \mathbb{R}^M$ is a sparse vector that selects a subset of time-location pairs. Most of its values are zero, leaving only a few time-location pairs that the individual focuses on. Discovering the sparsity patterns of $g^h$ based on the learned embedding is one of our major interests, as will be discussed shortly.

**Embedding of Time-Locations Pairs**    The inclusion of a time-location pair in the consideration set depends not only on its own features but also on its interactions with other time-location pairs. Consider, for example, a crime event scenario. A criminal's consideration of a specific time-location pair may be influenced by factors such as the time of day, police presence in the area, and the social environment. Moreover, the consideration of one time-location pair can affect others: if a thief considers a particular location for a potential crime, nearby locations may also be impacted.

To capture this complex interplay, we introduce a matrix $E^h \in \mathbb{R}^{M \times M}$ for each latent class $h \in [H]$. The diagonal elements of $E^h$ represent the self-influence of each time-location pair, while the off-diagonal elements represent the mutual influence between pairs. To encode the feature information of each time-location pair, we employ the following embedding

$$E^h = AW_A^h(BW_B^h)^\top \in \mathbb{R}^{M \times M}.$$

Here the matrices $A, B \in \mathbb{R}^{M \times M_0}$ encode the learnable embedded time-location information of all pairs shared by all latent classes. They are usually data-dependent, containing natural attributes of events, such as temporal and positional information, and demographics, in which case we denote them as $A^i, B^i$ for the $i$-th event. The matrices $W_A^h, W_B^h \in \mathbb{R}^{M_0 \times d}$ are linear transformations enabling the model to learn complex class-specific patterns.

**Learning Sparsity Patterns by Gating Function**    Based on the embedding of time-location pairs in the previous part, now we introduce a gating function to learn the sparsity pattern $g^h$.

To this end, let us denote
$$\boldsymbol{z}^h = AW_A^h(BW_B^h)^\top \mathbf{1} \in \mathbb{R}^M, \tag{6}$$

where $\mathbf{1}$ is a vector of ones. Then the $m$-th element of $\boldsymbol{z}$, denoted as $z_m^h$ representing the $m$-th row sum of $E^h$, can be interpreted as the aggregated influence score of all time-location pairs on the specific time-location pair $m$. This aggregated influence score will be used as input to the gate function, defined as
$$g^h = \alpha\text{-entmax}\,(\boldsymbol{z}^h) =: [(\alpha - 1)\boldsymbol{z}^h - \tau\mathbf{1}]_+^{1/\alpha-1},$$

that is, time-location pairs with aggregated influence score less than $\tau/(\alpha - 1)$ will be ruled out from the consideration set. Here $\tau, \alpha$ are hyper-parameters, and the $\alpha$-entmax function is a flexible family of transformations that generalizes the softmax function (Peters et al., 2019; Correia et al., 2019). The hyper-parameter $\alpha$ controls the sparsity of the output, with larger values of $\alpha$ leading to sparser outputs; the hyper-parameter $\tau$, which acts like a threshold, is the Lagrange multiplier corresponding to the $\sum_m g_m^h = 1$ constraint. The $\alpha$-entmax function is convex and differentiable, allowing for efficient optimization.

Our gating function serves as a soft, sparse ranking mechanism that selects a subset of time-location pairs based on their aggregated influence. The sparsity pattern is learned from the data and can adapt to different contexts. The gating function can be interpreted as a mechanism that captures the attention of individuals to different time-location pairs based on their features and interactions. By learning the sparsity pattern, the model can identify the most relevant time-location pairs for each latent class, providing insights into the decision-making process. Define a vector-valued function $f = [f_m]_{m=1,\dots,M}$ via

$$f_m(\boldsymbol{z}, \boldsymbol{u}) := \frac{\alpha\text{-entmax}_m(\boldsymbol{z})\exp(\boldsymbol{u}_m)}{\sum_{m'=1}^M \alpha\text{-entmax}_{m'}(\boldsymbol{z})\exp(\boldsymbol{u}_{m'})}. \tag{7}$$

Then, combining equations (5) (6) and (7), for the $i$-th event, our proposed spatial-temporal choice model predicts the likelihood of occurrence at the $m$-th time-location pair as

$$P_{im} = \sum_{h=1}^H \pi^h f_m\big(A^i W_A^h(B^i W_B^h)^\top \mathbf{1}, U\big), \quad m \in [M]. \tag{8}$$

As mentioned earlier, it is crucial to incorporate event-specific information into the matrices $A^i, B^i$, such as the characteristics of the individual who committed the crime and the type of event (e.g., robbery). This contextual information helps to ensure the probabilities for different events are appropriately calibrated.

**Likelihood function**    To represent the observed $N$ discrete events, we use a set of one-hot vectors $\{y_i\}_{i=1}^N$, where $y_i \in \mathbb{R}^M$ is a one-hot vector indicating if the $i$-th event takes place at the time-location pair $m$. Using (8), the log-likelihood function is given by:

$$\mathcal{L} = \sum_{i=1}^N \sum_{m=1}^M y_{im} \log P_{im}.$$

In the context of our counting process model, $y_{i,m}$ corresponds to the specific time and location where an event occurs. Thus, the likelihood function captures how well our model's predicted probabilities $P_{i,m}$ align with the observed occurrences of events at different time-location pairs. By optimizing this likelihood function, we refine our model parameters to accurately reflect the distribution of events based on the learned preferences and influences.

## 5    GENERALIZATION BOUND ANALYSIS

In this section, we analyze our proposed model's generalization capability when trained on a finite data set.

Suppose the $N$ events $\{(t_i, s_i)\}_{i=1}^N$–which can also be represented as $\{y_i\}_{i=1}^N$–are i.i.d. observations drawn from an underlying distribution $\mathcal{D}_*$. Each event is associated with an embedding $(A^i, B^i)$. Given these data, we learn the model (equation 8) using maximum log-likelihood estimation. Let $\mathcal{L}_{\mathcal{D}_*}(w)$ denote the expected loss associated with a parameter $w$ under the ground truth, and let $\mathcal{L}_N(\rho)$ denote the corresponding empirical loss with respect to the training data set. We are interested in bounding the generalization gap

$$\mathcal{L}_{\mathcal{D}_*}(\rho) - \mathcal{L}_{\hat{\mathcal{D}}_N}(\rho).$$

Using vector contraction inequality (Maurer, 2016), for a Lipschtiz loss, with high probability, the generalization gap is upper bounded by a multiple of the Rademacher complexity. Hence it suffices to bound the Rademacher complexity. To this end, we consider the following norm control on the parameters

$$\mathcal{W} := \left\{ w = \left\{ (W_A^h, W_B^h, U^h, \pi^h) \right\}_{h=1}^H : \right.$$

$$\left. \sum_{h=1}^H \pi_h \| W_A^h (W_B^h)^\top \|_F^2 \le C_W^2, \ \sum_{h=1}^H \pi_h \| U^h \|_F^2 \le C_U^2, \pi \in \Delta_H \right\}.$$

For a class of vector-valued functions defined in (8) with parameters in $\mathcal{W}$, we aim to derive an upper bound for the empirical Rademacher complexity with respect to the data set $\hat{\mathcal{D}}_N$:

$$\mathfrak{R}_n(\mathcal{W}) := \mathbb{E}_\epsilon \left[ \sup_{w \in \mathcal{W}} \frac{1}{N} \sum_{n=1}^N \sum_{m=1}^M \epsilon_{nm} \sum_{h=1}^H \pi^h f_m \big( A^i W_A^h (B^i W_B^h)^\top \mathbf{1}, U^h \big) \right].$$

We have the following result.

**Theorem 1.** *Suppose the Frobenius norm of all embedding matrices $A^i$, $B^i$ are bounded by $\nu$, and the maximal cardinality of the consideration set is $\kappa$. Let $L$ be the Lipschitz constant of the function $f$ as defined in (7). Then it holds that*

$$\mathfrak{R}_{\hat{\mathcal{D}}_N}(\mathcal{W}) \le \frac{\sqrt{2\kappa} L (\nu^2 C_W + \sqrt{M} C_U)}{\sqrt{N}}.$$

Theorem 1 demonstrates that the generalization bound is of order $O(1/\sqrt{N})$. Notably, this bound is independent of the number of latent classes, $H$, regardless of the mixture distribution $\pi$, which is a desirable feature of the proposed model. The two components of the constants in the generalization bound arise from the complexities of the embedding matrices $\{(W_A^h, W_B^h)\}_{h=1}^H$ and the utilities $\{U^h\}_{h=1}^H$, respectively, both of which exhibit a mild dependence on the controlled norm. The first component also depends on the maximal size of the consideration set, which is typically small. The second component includes an additional square-root factor that depends on the number of time-location pairs, due to the fact that the utility for each pair is treated separately. In practice, these pairs often have a lower-dimensional parameterization. In this case, it is easy to obtain a more favorable constant using such dimension reduction.

# 6 EXPERIMENTS

To assess the efficacy of our proposed framework, we initially apply our model to three real-world spatial-temporal datasets. We demonstrate that our model accurately aligns with actual event probabilities and reveals distinct preferences in mined expert patterns. Subsequently, we evaluate our model's predictive performance against existing methods, revealing its superior accuracy compared to baseline models. Moreover, our approach introduces a novel viewpoint in spatial-temporal modeling. Initially fitting a Log-Gaussian Cox process model, we then employ our choice model for further explanation. In comparison to traditional non-negative matrix factorization, our model exhibits enhanced interpretability.

## 6.1 EXPERIMENTAL SETUP

**Datasets**   We considered three real-world spatial-temporal datasets. Followings are brief introduction to these datasets: *i) New York Crime*[1]: This dataset includes all valid felony, misdemeanor, and violation crimes reported to the New York City Police Department (NYPD) for all complete quarters. We extracted records from January 1, 2024, totaling 732 crime incidents. The date time was divided into four time slots, and the New York area was segmented into 100 small area blocks based on longitude and latitude. *ii) Chicago Crime*[2]: This dataset reflects reported incidents of crime (with the exception of murders where data exists for each victim) that occurred in the City of Chicago from 2001 to present. Data is extracted from the Chicago Police Department's CLEAR (Citizen Law Enforcement Analysis and Reporting) system. We extracted records from the Chicago area on July 5, 2024, totaling 861 crime incidents. Similarly, the date time was segmented into four time slots, and the Chicago area was divided into 100 small area blocks based on longitude and latitude. *iii) Shanghai Mobike*[3]: This dataset documents bicycle-sharing rental events in Shanghai during August 2016. We extracted records from the Shanghai area on August 7, 2016, totaling 2095 Mobike usage events. Considering potential variations in bike-sharing patterns during peak hours on workdays, we divided the day into six time slots, with 8:00 am to 12:00 pm and 16:00 pm to 20:00 pm specifically encompassing morning and evening rush hours. Similarly, we segmented the Shanghai area into 100 small area blocks based on longitude and latitude.

For all datasets, the date we choose to focus on is randomly selected. Additionally, for prediction tasks, we extracted records from each of the three datasets corresponding to the day immediately following the dates in the current dataset, enabling model comparison when forecasting future events.

**Baselines**   To evaluate the capability of our proposed models, we compare against commonly-used baselines and state-of-the-art models. *i) ARMA* (Araghinejad & Araghinejad, 2014): Auto-Regression-Moving-Average is well-known for predicting time series data. ARMA predicts the event number of a region solely based on the historical event records of the region, considering the recent time slots for a moving average. *ii) CSI* (de, 1978): Cubic Spline Interpolation trains piecewise third-order polynomials which pass through event points of recent time slots, and then predicts the event number in the near future by the trained polynomials. *iii) LGCP* (Diggle et al., 2013; Miller et al., 2014): Log-Gaussian Cox Process is a kind of Poisson process with varying intensity, where the log-intensity is assumed to be drawn from a Gaussian process. *iv) NSTPP* (Chen et al., 2020): It applies Neural ODEs as the backbone, which parameterized the temporal intensity with Neural Jump SDEs and the spatial PDF with continuous-time normalizing flows. *v) DSTPP* (Yuan et al., 2023): it leverages diffusion models to learn complex spatio-temporal joint distributions. *vi) ST-HSL* (Li et al., 2022): It proposes a Spatial-Temporal Self-Supervised Hypergraph Learning framework for crime prediction.

**Comparison Metric**   We evaluate the prediction performance of models using:

*i) aRMSE*: For $N = 100$ disjointed area blocks for all three datasets, in each block, we use the data of previous $K$ time slots to train the models and predict the event number of $K_S$ time slots later. Therefore, the event prediction performance is evaluated in terms of the average root-mean-square-

---

[1]https://data.cityofnewyork.us/Public-Safety/NYPD-Complaint-Data-Current-Year-To-Date-/5uac-w243

[2]https://data.cityofchicago.org/Public-Safety/Crimes-2001-to-Present/ijzp-q8t2

[3]https://github.com/Andrehinh/Interesting-python/tree/master/Mobike

error (*RMSE*) of all $N$ blocks as:

$$aRMSE = \frac{1}{N} \sum_{n=1}^{N} \sqrt{\frac{1}{K_S} \sum_{k=1}^{K_S} \left(\hat{Y}_n^k - Y_n^k\right)^2} \tag{9}$$

where $\hat{Y}_n^k$ is the predicted event number, while $Y_n^k$ is the observed event number. In this evaluation, we choose $K = K_S = 4$ for New York and Chicago Crime dataset, and $K = K_S = 6$ for mobike dataset.

*ii) MAPE*: It is also used to evaluate the event number prediction but is more interpretable in terms of the absolute error in predictions. It is defined as the absolute difference between predicted number of events and actual number:

$$MAPE = \sum_{n=1}^{N} \sum_{k=1}^{K_S} \frac{|\hat{Y}_n^k - Y_n^k|}{Y_n^k} \tag{10}$$

Where the notation used here aligns with those of *aRMSE*.

## 6.2  FINDINGS OF OUR MODEL

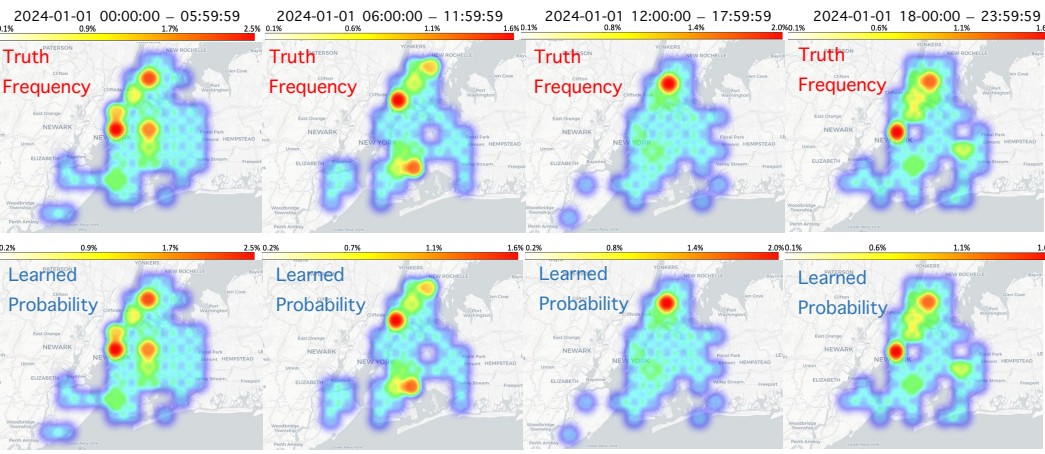

Figure 2: Comparison of the actual crime frequency and the modeled probability on January 1, 2024, in New York City.

As shown in Fig. 2 and Tab. 1, our model closely aligns its fitted probabilities with actual frequency. The model identified the time-location pair with the highest probability, 0.0245, at a small area centered by University PL (40.731, -73.995), within the time slot 00:00:00 to 05:59:59, matching the true frequency of 0.0246. Across the top 10 time-location pairs, our model's fitted probabilities consistently align with the ground truth frequencies, with slight discrepancies. The highlighted areas in Fig. 2 indicate regions where the probability of crime events occurrence is higher at a specific time duration. For instance, during the early hours of the morning (00:00:00-05:59:59), the likelihood of crime events is higher in three areas centered around Monterery Avenue (40.731, -73.995), Bronx Food Stamps Office (40.847, -73.895), and 53 AVE (40.731, -73.895).

Furthermore, our model can offer a novel perspective on explaining the fitted probabilities. In this experiment, we encode the severity of the crime, as well as the race and age of the suspect, as learnable embeddings. Visualization of the learned utility function can be found in Appendix. C. Here we present different expert-pattern for each time-location pair and its corresponding mixing coefficient in Fig. 3. Seeing the results, expert-1 plays a predominant role, as indicated by the highest mixing coefficient. The time-location patterns discovered by expert-1 closely resemble the fitted probability patterns. We want to emphasize that the crime time-location patterns uncovered by expert-2 and expert-3 are also noteworthy, revealing crime tendencies that are often overlooked. For example, expert-2 identified that during the time slot 12:00:00-17:59:59, the area near Oak Point Ave (40.808, -73.895) and Pulaski St (40.693,-73.945) also exhibits higher crime probabilities,

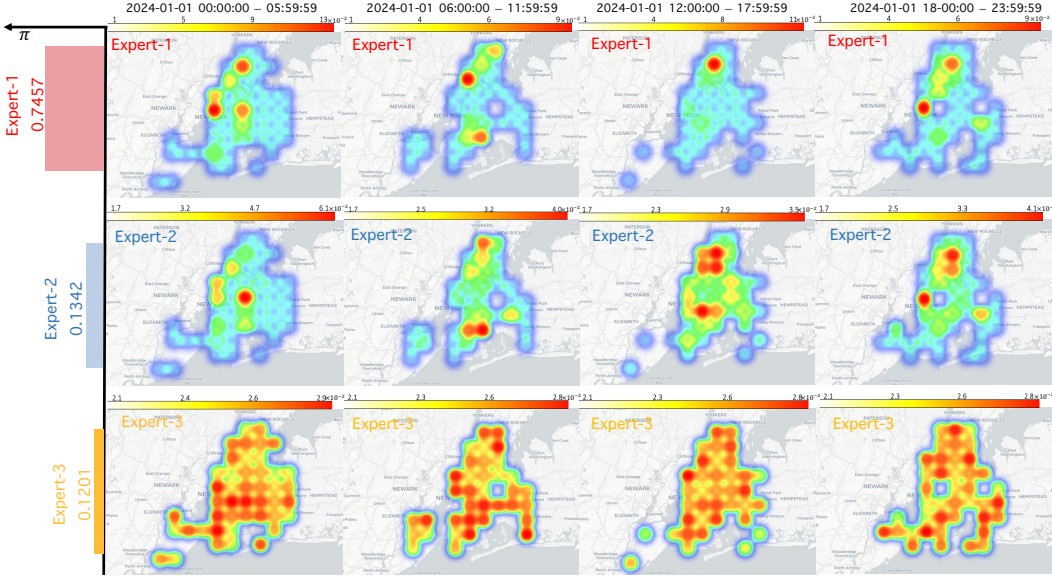

Figure 3: Mixing coefficient $\pi^h$ (Left bar plot) and mixture pattern adjusted by utility score $(g^h \exp(U^h))$ for different experts (Right heatmaps) on January 1, 2024, in New York City. The selection of the number of experts is based on empirical experiments.

which are ignored by expert-1. Oak Point Ave and Pulaski St are located in the Bronx. This area is characterized by a challenging economic landscape, with high unemployment rates. Additionally, the neighborhood faces issues with youth involvement in gangs, particularly during after-school hours.

We conducted the same experiment on other datasets, and the results are presented in Appendix.C.

| Top-10 Ground Truth Frequency | | | Top-10 Learned probability | | |
|---|---|---|---|---|---|
| Time slot | (Lat., Lon.) | Freq. | Time slot | (Lat., Lon.) | Prob. |
| 00:00:00-05:59:59 | (40.731, -73.995) | 0.0246 | 00:00:00-05:59:59 | (40.731, -73.995) | 0.0245 |
| 00:00:00-05:59:59 | (40.847, -73.895) | 0.0232 | 00:00:00-05:59:59 | (40.847, -73.895) | 0.0231 |
| 00:00:00-05:59:59 | (40.731, -73.895) | 0.0219 | 00:00:00-05:59:59 | (40.731, -73.895) | 0.0218 |
| 12:00:00-17:59:59 | (40.847, -73.895) | 0.0205 | 12:00:00-17:59:59 | (40.847, -73.895) | 0.0204 |
| 00:00:00-05:59:59 | (40.770, -73.995) | 0.0205 | 00:00:00-05:59:59 | (40.770, -73.995) | 0.0204 |
| 00:00:00-05:59:59 | (40.808, -73.945) | 0.0191 | 00:00:00-05:59:59 | (40.808, -73.945) | 0.0192 |
| 00:00:00-05:59:59 | (40.693, -73.895) | 0.0178 | 00:00:00-05:59:59 | (40.693, -73.895) | 0.0178 |
| 06:00:00-11:59:59 | (40.808, -73.945) | 0.0164 | 06:00:00-11:59:59 | (40.808, -73.945) | 0.0163 |
| 18:00:00-23:59:59 | (40.731, -73.995) | 0.0164 | 00:00:00-05:59:59 | (40.615, -73.995) | 0.0163 |
| 00:00:00-05:59:59 | (40.615, -73.995) | 0.0164 | 18:00:00-23:59:59 | (40.731, -73.995) | 0.0163 |

Table 1: Ground truth crime frequency and learned probability for top-10 time-location pair on January 1, 2024, in New York City. The red font indicates a match between the ranks of frequency and probability of a time-location pair.

## 6.3 EXPERIMENTS FOR PREDICTION TASKS

To enable predictive functionality based on the probabilities of events at each time-location, our choice model requires prior statistical knowledge. In our experiments, we predict the number of events that may occur the next day at each time-location pair by multiplying the fitted probabilities with the average events count in ten days prior to the targeted prediction date. The results in Tab.2 demonstrate that our model surpasses all baseline methods in terms of prediction accuracy.

## 6.4 EXPERIMENTS FOR EXPLAINING OTHER MODELS

Our model offers an alternative perspective to other spatial-temporal models. We first fit a Log-Gaussian Cox Process (LGCP) on New Your Crime dataset, which is a doubly-stochastic Poisson

| Model | NYC Crime | | Chicago Crime | | Shanghai Mobike | |
|---|---|---|---|---|---|---|
| | aRMSE ↓ | MAPE ↓ | aRMSE ↓ | MAPE ↓ | aRMSE ↓ | MAPE ↓ |
| AMAR | 4.95 +/- 0.27 | 120.57 +/- 4.23 | 6.63 +/- 0.52 | 132.17 +/- 4.58 | 6.84 +/- 0.18 | 178.63 +/- 6.10 |
| CSI | 5.24 +/- 0.13 | 118.34 +/- 3.28 | 6.41 +/- 0.13 | 124.67 +/- 4.12 | 6.21 +/- 0.23 | 182.82 +/- 6.73 |
| LGCP | 4.90 +/- 0.46 | 108.09 +/- 2.50 | 6.32 +/- 0.33 | 151.52 +/- 2.82 | 7.62 +/- 0.53 | 195.70 +/- 2.62 |
| NSTPP | 3.90 +/- 0.33 | 115.98 +/- 2.70 | 5.45 +/- 0.67 | 122.08 +/- 3.05 | 7.30 +/- 0.72 | 160.18 +/- 4.33 |
| DSTPP | 3.82 +/- 0.13 | 112.47 +/- 3.47 | 5.27 +/- 0.28 | 120.23 +/- 3.50 | 6.42 +/- 0.82 | 173.46 +/- 3.58 |
| ST-HSL | 4.47 +/- 0.37 | 163.33 +/- 5.67 | 5.68 +/- 0.46 | 119.96 +/- 3.61 | 15.29 +/- 0.97 | 226.51 +/-10.21 |
| **Ours*** | **2.34 +/- 0.05** | **107.94 +/- 4.27** | **2.19 +/- 0.04** | **87.38 +/- 3.12** | **3.28 +/- 0.04** | **154.62 +/- 7.05** |

Table 2: Comparison between our model and baselines for prediction tasks. Bold text represents the best result. The performance is averaged over three different seeds and the standard deviation is stored after "$+/-$".

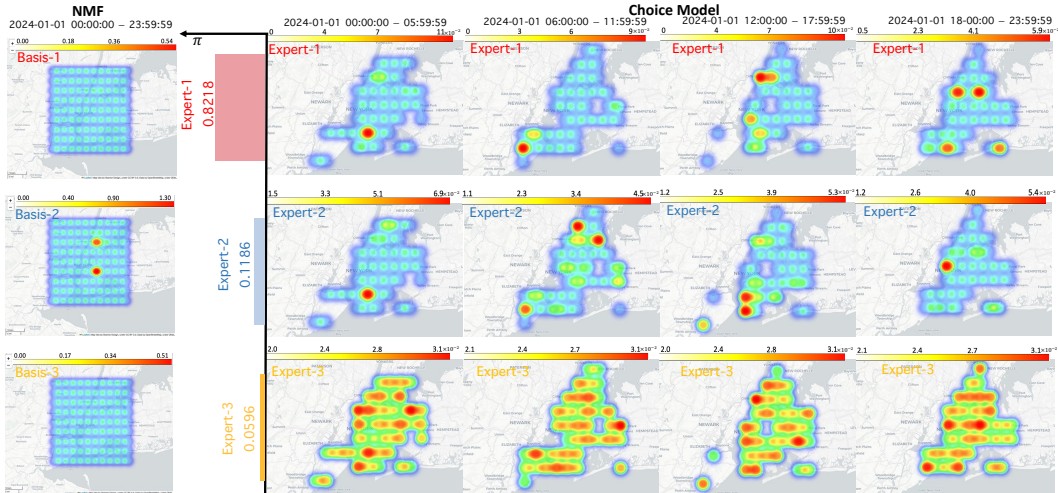

Figure 4: Comparison of the learned expert pattern of our choice model and the basis of non-negative matrix factorization (NMF) on New York City crime dataset. Left: NMF basis, Right: expert pattern of our choice model

process with a spatially varying intensity function modeled as an exponentiated Gaussian Process

$$Z(\cdot) \sim GP(0, k(\cdot, \cdot)), \quad \lambda(\cdot) \sim \exp(Z(\cdot)), \quad x_1, ..., x_N \sim PP(\lambda(\cdot)) \tag{11}$$

where $GP$ refer to a Gaussian Process, $PP$ refer to a Poisson process. $k(\cdot, \cdot)$ represents the squared exponential covariance function and $x_i$ represents a countable collection of independent Poisson process with measure $\lambda_i$. Fig. 4 exhibits two ways to explain the results from LGCP. Non-negative matrix factorization (Lee & Seung, 1999; Miller et al., 2014) captures few information in different bases. Our model offers a more granular explanation at the level of time-location pairs. The details of how to interpret LGCP results with our model and NMF are presented in Appendix. C.

## 7 CONCLUSION

We propose a new approach to model spatial-temporal counting processes that effectively captures the complexity of human decision-making. By using a mixture model, our method learns diverse sparse patterns representing different underlying preferences and decision strategies, accounting for heterogeneous behaviors in the data. The gating function automatically identifies these sparse patterns, while an additional utility function adds flexibility, ensuring adaptability across various scenarios. The asymmetry in the matrix $H$ highlights the non-reciprocal nature of mutual influences, crucial for modeling directional relationships between time-location pairs. Our approach not only addresses the curse of dimensionality but also enhances interpretability, offering a clear understanding of individual behaviors in time and location contexts.

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

APPENDIX OVERVIEW

In the following, we will provide supplementary materials to better illustrate our methods and experiments.

## A  ALGORITHM DETAILS

### A.1  SPATIAL-TEMPORAL EMBEDDING

We adopt a spatial-temporal embedding method akin to that described in (Alexey, 2020; Aksan et al., 2021). Further details can be found in Fig.1. Initially, the region is segmented into distinct blocks. To encode block order, we introduce sinusoidal positional encoding for area position embedding. Subsequently, linear embedding is utilized for spatial information, typically latitude and longitude. Temporal information is encoded using sinusoidal positional encoding as described in (Zuo et al., 2020). In the context of decision-making, relevant static features can be encoded using one-hot semantic encoding, while dynamic features are encoded linearly. The embeddings for spatial, temporal, and relevant feature information are then combined via element-wise addition to generate the comprehensive embedding.

First, we divide the area into disjoint blocks. To inject a notation of block ordering we add sinusoidal positional encoding for these area position embedding. Then considering the spatial information which are usually latitude and longitude, we apply linear embedding. For encoding temporal information, we also adopt sinusoidal positional encoding a (Zuo et al., 2020). Considering other relevant features related to the decision-making process, we can apply one-hot semantic encoding for static feature and linear encoding for dynamic features. Finally, the embedding for spatial information, temporal information, and relevant feature information then directly element-wise addition together to obtain the overall embedding.

This approach is similar to the use of positional embeddings and feature embeddings in attention mechanisms, where initial embeddings are transformed through linear projections to capture more nuanced information. By combining the base embeddings $A$ and $B$ with these flexible projections, our model can more accurately represent and adapt to diverse preference patterns and social influences, enriching the overall decision-making framework.

### A.2  OVERALL ALGORITHM

## B  THEORETICAL DETAILS

### B.1  DETAILS OF $\alpha$-ENTMAX

$$\alpha\text{-entmax}\,(\boldsymbol{z}) := \operatorname*{argmax}_{\boldsymbol{p}\in\Delta^M}\quad \boldsymbol{p}^\top\boldsymbol{z} + \mathrm{H}_\alpha^{\mathrm{T}}(\boldsymbol{p}), \tag{12}$$

where $\triangle^M := \left\{\boldsymbol{p}\in\mathbb{R}^M : \sum_i p_i = 1\right\}$ is the probability simplex, and, for $\alpha \geq 1$, $\mathrm{H}_\alpha^{\mathrm{T}}$ is the Tsallis continuous family of entropies (Tsallis, 1988):

$$\mathrm{H}_\alpha^{\mathrm{T}}(\boldsymbol{p}) := \begin{cases} \frac{1}{\alpha(\alpha-1)}\sum_j \left(p_j - p_j^\alpha\right), & \alpha \neq 1 \\ -\sum_j p_j \log p_j, & \alpha = 1 \end{cases}$$

**Algorithm 1** Learning the Model Parameters for the Mixture-of-Experts Model

---

1: **Input:** Observed data $\{y_{i,m}\}_{i=1}^{N}$, initial parameters $\boldsymbol{\theta} = [\pi, A, B, \{[\alpha^h, \tau^h, W_h^a, W_h^b, U_h]\}_{h \in [H]}]$

2: **Output:** Optimized model parameters $\boldsymbol{\theta}^*$

3: **Initialization:** Initialize $\boldsymbol{\theta}$ randomly or heuristically.

4: **Description:** $A$ and $B$ serve as shared feature embeddings that encode the positional and contextual information necessary for understanding the preference distribution in generating the events.

5: **for** each expert $h \in [H]$ **do**

6:     Compute gating function:

$$g^h = \mathbf{g}_{\alpha^h, \tau^h} \left( A^h \left( B^h \right)^\top \mathbf{1} \right),$$

    where $A^h = A W_a^h$ and $B^h = B W_b^h$.

7: **end for**

8: **for** each event $i \in [N]$ and each pair $m \in [M]$ **do**

9:     Compute probability:

$$P_m = \sum_{h=1}^{H} \pi^h \frac{g_m^h \exp \left( U_m^h \right)}{\sum_{m'=1}^{M} g_{m'}^h \exp \left( U_{m'}^h \right)}.$$

10: **end for**

11: **Optimize:** Maximize the likelihood function:

$$\mathcal{L}(\boldsymbol{\theta}) = \prod_{i=1}^{N} \prod_{m=1}^{M} \left( P_{i,m} \right)^{y_{i,m}}$$

    to update $\boldsymbol{\theta}$ using gradient descent or a similar optimization method.

12: **Output:** Optimized parameters $\boldsymbol{\theta}^*$.

---

This family contains the well-known Shannon and Gini entropies, corresponding to the cases $\alpha = 1$ and $\alpha = 2$, respectively.

### B.2 PROOF OF THEOREM 1

**Lemma 1.** $\alpha$-entmax$_m(\boldsymbol{z})$ is 1-Lipschitz continuous w.r.t. $l_2$ norm.

*Proof.* Using the variational representation of $\alpha$-entmax,

$$\alpha\text{-entmax}(\boldsymbol{z}) = \underset{\boldsymbol{p} \in \Delta^M}{\operatorname{argmax}} \quad \boldsymbol{p}^\top \boldsymbol{z} + \mathrm{H}_\alpha^\mathrm{T}(\boldsymbol{p})$$

By the Envelope theorem, its subgradient belongs to $\Delta^M$, and hence is bounded by $\max_{\mathbf{p} \in \Delta^M} ||\mathbf{p}||_2 = 1$. As a result, $\alpha$-entmax is 1-Lipschitz continuous in $l_2$-norm. $\square$

**Lemma 2.** *Define a function*

$$f_m(\boldsymbol{z}, \boldsymbol{u}) := \frac{\alpha\text{-entmax}_m(\boldsymbol{z}) \exp(\boldsymbol{u}_m)}{\sum_{m'=1}^M \alpha\text{-entmax}_{m'}(\boldsymbol{z}) \exp(\boldsymbol{u}_{m'})}.$$

*Then $f_m$ is L-Lipschitz, where $L := \frac{1}{l}$. $l$ is the minimum value of positive $\alpha$-entmax$_m(\boldsymbol{z})$.*

*Proof.* Note that $\alpha$-entmax$(\boldsymbol{z})$ is a sparse gating function, with a set of $[M_+] \subset [M]$ non-zero items. We consider $\alpha$-entmax$(\boldsymbol{z}) > 0$ and $\alpha$-entmax$_m(\boldsymbol{z}) = 0$ separately.

(i) For $m$ such that $\alpha$-entmax$_m(\boldsymbol{z}) > 0$,
$f_m(\boldsymbol{z}, \boldsymbol{u})$ can be writen as softmax$_m(\log(\alpha\text{-entmax}(\boldsymbol{z})) + \boldsymbol{u})$. By lemma 1, we can derive the Lipschitz constant of $\log(\alpha\text{-entmax}_m(\boldsymbol{z}))$.

$$\nabla log(\alpha\text{-entmax}_m(\boldsymbol{z})) = \frac{\nabla \alpha\text{-entmax}_m(\boldsymbol{z})}{\alpha\text{-entmax}_m(\boldsymbol{z})} \leq \frac{\nabla \alpha\text{-entmax}_m(\boldsymbol{z})}{\min(\alpha\text{-entmax}_m(\boldsymbol{z}))}$$

Let's denote $\min(\alpha\text{-entmax}_m(\boldsymbol{z}))$ as $l$. Then $\log(\alpha\text{-entmax}_m(\boldsymbol{z}))$ is $\frac{1}{l}$-Lipschitz. Adding $U$ does not change the Lipschitz constant of a function. Since $softmax(\cdot)$ is a 1-Lipschitz function, the composite function softmax$_m(\log(\alpha\text{-entmax}(\boldsymbol{z})) + \boldsymbol{u})$ is also $\frac{1}{l}$-Lipschitz.

(ii) For $m$ such that $\alpha$-entmax$_m(\boldsymbol{z}) = 0$,
we consider another $\alpha$-entmax$_m(\boldsymbol{z'})$. If $\alpha$-entmax$_m(\boldsymbol{z'}) = 0$, $\frac{1}{l}$-Lipschitz also holds. If $\alpha$-entmax$_m(\boldsymbol{z'}) > 0$, by Mean Value Theorem, there exits $\alpha$-entmax$_m(\boldsymbol{z''}) > 0, \boldsymbol{z''} \in [\boldsymbol{z}, \boldsymbol{z'}]$, where previous analysis holds.

$\square$

With the above definition of $f_m$, for a sample $(A^i, B^i)$, the choice probability of option $m$ for the latent class $h$ can be written as

$$\sigma_m\left(A^i, B^i; W_A^h, W_B^h, U^h\right) = f_m\left(\{A_j^n W_A^h (B^i W_B^h)^\top \mathbf{1}\}_{j=1,\dots,M}, U^h\right),$$

where $A_j^n$ denotes the $j$-th row of the matrix $A^i$.

Let us derive a bound for the empirical Rademacher complexity. Since a linear functional of the probability distribution $\pi$ attains its supremum at the point mass, the above expectation equals

$$\mathbb{E}_\epsilon \left[ \sup_{||W_A W_B^\top||_F \leq C_W, ||U||_F \leq C_U} \frac{1}{N} \sum_{n=1}^N \sum_{m \in \mathcal{M}_n} f_m\left(A^i W_A (B^i W_B)^\top \mathbf{1}, U\right) \right],$$

where $\mathcal{M}_n$ be the subset of $[M]$ for which $f_m \neq 0$. Since $f_m$ is $L$-Lipschitz, using the vector contraction lemma (Maurer, 2016), it holds that

$$\mathfrak{R}_n(\mathcal{W}) \leq \sqrt{2}L \cdot \mathbb{E}_\epsilon \left[ \sup_{\boldsymbol{w} \in \mathcal{W}} \frac{1}{N} \sum_{n=1}^N \sum_{m \in \mathcal{M}_n} \sum_{j=1}^M \left( \epsilon_{nmj}^W A_j^n W_A (B^i W_B)^\top \mathbf{1} + \epsilon_{nmj}^U U_j \right) \right].$$

By the additive separability of the Rademacher complexity, the expectation above equals

$$
\mathbb{E}_\epsilon \left[ \sup_{\|W_A W_B^\top\|_F \leq C_W} \frac{1}{N} \sum_{n=1}^N \sum_{m \in \mathcal{M}_n} \sum_{j=1}^M \epsilon_{nmj}^W A_j^n W_A (B^i W_B)^\top \mathbf{1} \right] + \mathbb{E}_\epsilon \left[ \sup_{\|U\|_2 \leq C_U} \frac{1}{N} \sum_{n=1}^N \sum_{m \in \mathcal{M}_n} \sum_{j=1}^M \epsilon_{nmj}^U U_j^h \right].
$$

$$(13)$$

For the first term, we rewrite it as

$$
\sup_{\|W_A W_B^\top\|_F \leq C_W} \left\langle W_A W_B^\top, \; \frac{1}{N} \sum_{n=1}^N \sum_{m \in \mathcal{M}_n} \sum_{j=1}^M \epsilon_{nmj}^W (B^i)^\top \mathbf{1} A_j^n \right\rangle.
$$

which, by Cauchy-Schwarz inequality, is upper bounded by

$$
\frac{C_W}{N} \left\| \sum_{n=1}^N \sum_{m \in \mathcal{M}_n} \sum_{j=1}^M \epsilon_{nmj}^W (B^i)^\top \mathbf{1} A_j^n \right\|_F.
$$

By Jensen's inequality,

$$
\mathbb{E}_\epsilon \left[ \left\| \sum_{n=1}^N \sum_{m \in \mathcal{M}_n} \sum_{j=1}^M \epsilon_{nmj}^W (B^i)^\top \mathbf{1} A_j^n \right\|_F \right] \leq \sqrt{ \mathbb{E}_\epsilon \left[ \left\| \sum_{n=1}^N \sum_{m \in \mathcal{M}_n} \sum_{j=1}^M \epsilon_{nmj}^W (B^i)^\top \mathbf{1} A_j^n \right\|_F^2 \right] }
$$

$$
\leq \sqrt{ \sum_{n=1}^N \sum_{m \in \mathcal{M}_n} \sum_{j=1}^M \| (B^i)^\top \mathbf{1} A_j^n \|_F^2 }
$$

$$
\leq \sqrt{ \kappa \sum_{n=1}^N \sum_{j=1}^M \| (B^i)^\top \mathbf{1} A_j^n \|_F^2 }
$$

Observe that

$$
\sum_{j=1}^M \| (B^i)^\top \mathbf{1} A_j^n \|_F^2 = \| (B^i)^\top \mathbf{1} \|_2^2 \cdot \sum_{j=1}^M \| A_j^n \|_2^2 \leq \| B^i \|_F \cdot \| A^i \|_F.
$$

Hence, the first term of (13) is bounded by

$$
\frac{\sqrt{\kappa} \nu^2 C_W}{\sqrt{N}}.
$$

Similarly, we can show that the second term of (13) is bounded by

$$
\mathbb{E}_\epsilon \left[ \sup_{\|U\|_2 \leq C_U} \frac{1}{N} \sum_{n=1}^N \sum_{m \in \mathcal{M}_n} \sum_{j=1}^M \epsilon_{nmj}^U U_j \right] \leq \frac{C_U}{N} \mathbb{E}_\epsilon \left[ \left\| \sum_{n=1}^N \sum_{m \in \mathcal{M}_n} \epsilon_{nm\cdot}^U \right\|_2 \right]
$$

$$
\leq \frac{\sqrt{\kappa M} C_U}{\sqrt{N}}.
$$

## C  MORE EXPERIMENT RESULTS

### C.1  LEARNED UTILITY FUNCTION ON NYC CRIME DATASET

The results are shown in Fig.5. Seeing from the value, the learn utility function play an important role in adjusting the choices based on social characteristics. The patterns of $U$ scores are similar with the mixture patterns adjusted by utility scores.

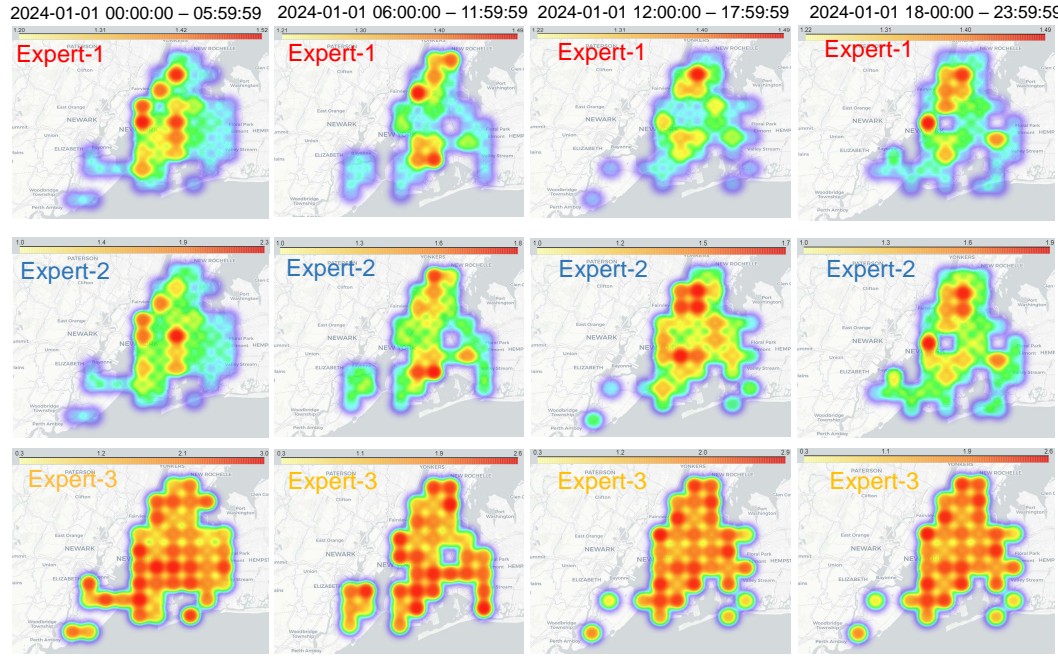

Figure 5: Learned $U$ function for each expert in the experiment January 1, 2024, in New York City.

## C.2 HOW TO EXPLAIN RESULTS OF LGCP

We fit our model using a new objective function based on the least squared error between estimated probability and the probability from the LGCP. This approach allows us to interpret the expert patterns learned by our model to explain the already fitted LGCP model.

We conduct non-negative matrix factorization on the learned intensity matrix $\Lambda \in \mathbb{R}^{T \times S}$ from LGCP.

$$\Lambda \approx WB$$

where $W \in \mathbb{R}^{T \times H}$ is the weight matrix, and $B \in \mathbb{H}^{T \times S}$ is the basis matrix. $S$ is the number of spatial grids, and $T$ is the number of temporal intervals. $H$ is the number of mixture, which is set to be same as our model.

## C.3 FIGURES

We fit our model on two other real-world spatial-temporal datasets. Fig.6 and Fig.7 are the results on Chicago Crime dataset. Fig.8 and Fig.9 are the results on Shanghai mobike renting dataset.

## D COMPUTING INFRASTRUCTURE

All synthetic data experiments, as well as the real-world data experiments, including the comparison experiments with baselines, are performed on Ubuntu 20.04.3 LTS system with Intel(R) Xeon(R) Gold 6248R CPU @ 3.00GHz, 227 Gigabyte memory.

## E SCALABILITY

Across all experiments, as the dataset sample size increases, both evaluation metrics, aRMSE and MAPE, decrease for the prediction task. Taking the NYC dataset as an example, with an increase in data size from the current 732 samples to 5561 samples, aRMSE decreases to 2.06, and MAPE decreases to 100.72. The training time required for model convergence remains within acceptable limits on the current computing infrastructure. For the NYC dataset with 5561 samples, the model converges in only 3.8970 hours. Even for the large-scale Mobike dataset with 8786 samples, our model converges and achieves good inference and prediction results after training for approximately 7.1056 hours.

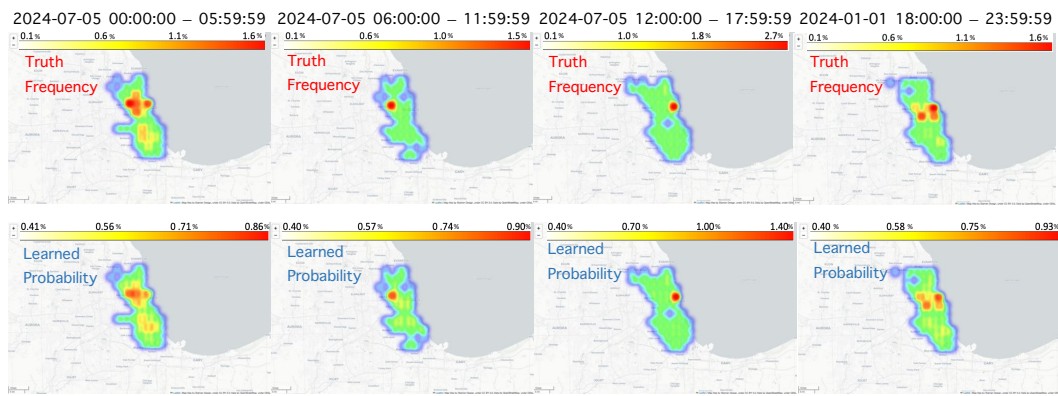

Figure 6: Comparison of the actual crime frequency and the modeled probability on July 5, 2024, in Chicago.

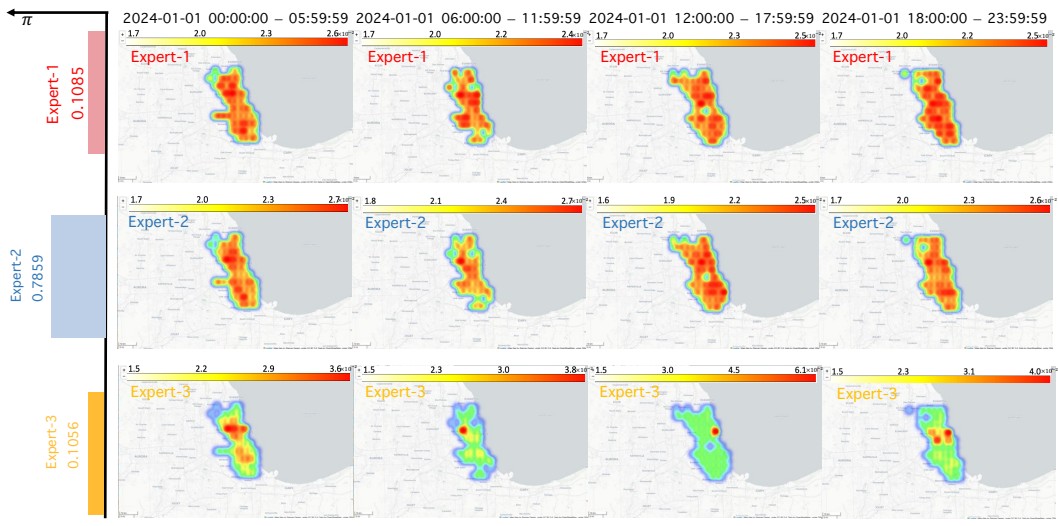

Figure 7: Mixing coefficient $\pi^h$ (Left bar plot) and mixture pattern adjusted by utility score $(g^h \exp(U^h))$ for different experts (Right heatmaps) on July 5, 2024, in Chicago. The selection of the number of experts is based on empirical experiments.

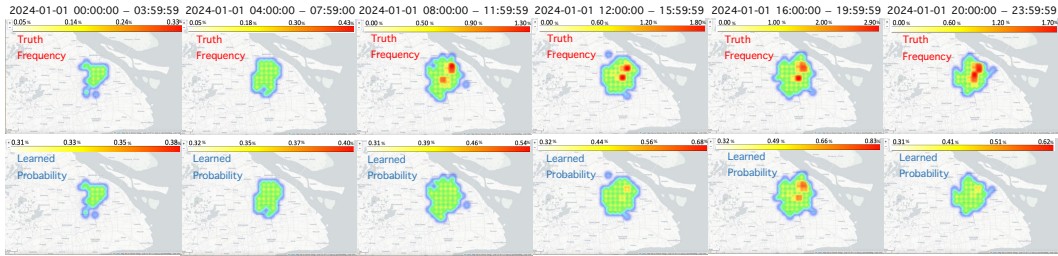

Figure 8: Comparison of the actual mobike renting frequency and the modeled probability on August 7, 2016, in Shanghai.

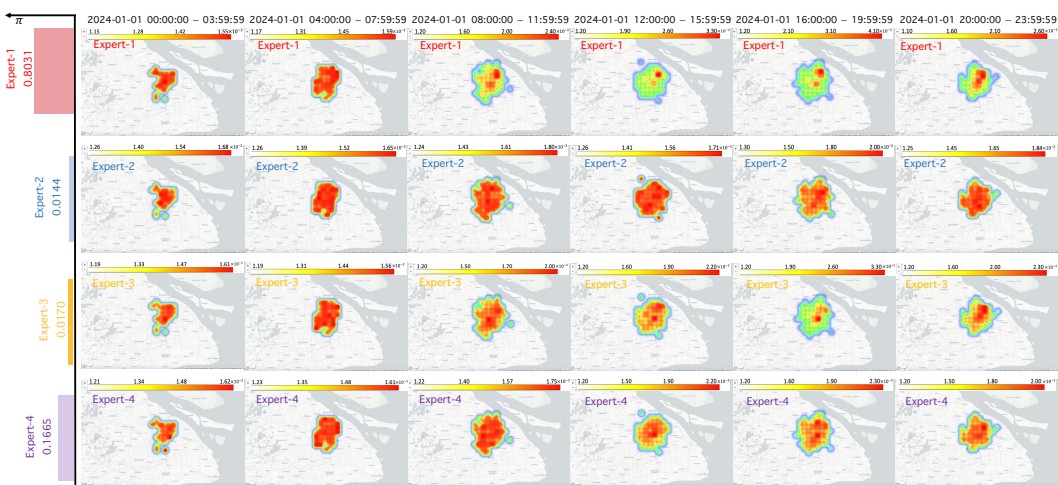

Figure 9: Mixing coefficient $\pi^h$ (Left bar plot) and mixture pattern adjusted by utility score $(g^h \exp(U^h))$ for different experts (Right heatmaps) on August 7, 2016, in Shanghai. The selection of the number of experts is based on empirical experiments.

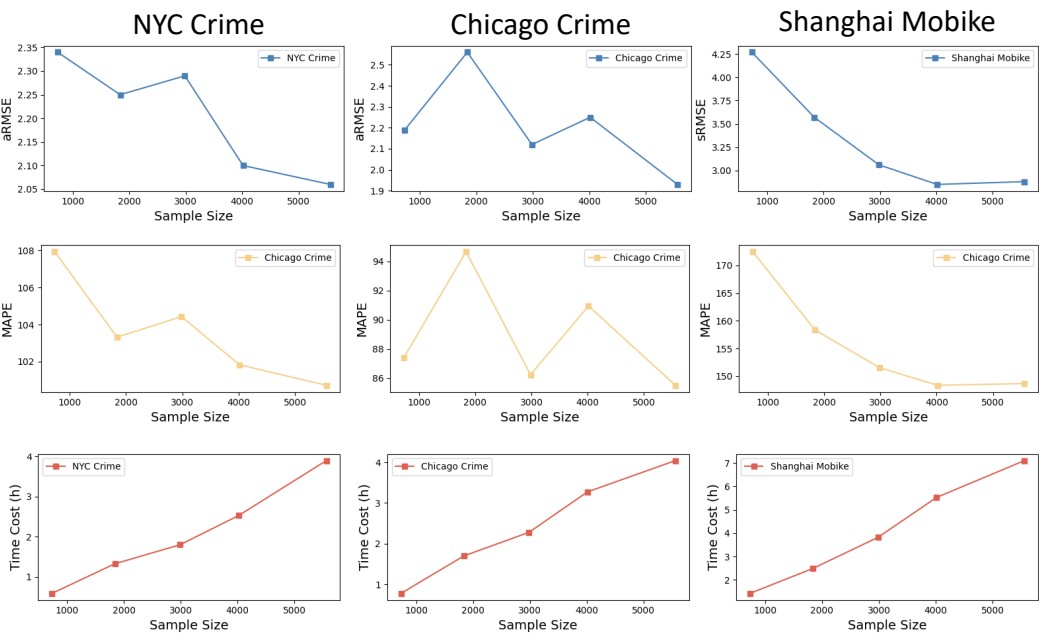

Figure 10: Scalability and the computation time cost of our method. For each dataset, we vary the sample size from small to large scale. Specifically, for NYC Crime dataset, we extract the records from 2024-01-01 – 2024-01-01: 732 samples, 2024-01-01 – 2024-01-03: 1840 samples, 2024-01-01 – 2024-01-05: 2985 samples, 2024-01-01 – 2024-01-07: 4016 samples, 2024-01-01 – 2024-01-10: 5561 samples. For Chicago Crime dataset, we extract the records from 2024-07-05 – 2024-07-05: 861 samples, 2024-07-05 – 2024-07-07: 2434 samples, 2024-07-05 – 2024-07-08: 3207 samples, 2024-07-05 – 2024-07-10: 4578 samples, 2024-07-05 – 2024-07-11: 5321 samples. For Shanghai Mobike dataset, we extract the records from 2016-08-02 – 2016-08-02: 1457 samples, 2016-08-02 – 2016-08-03: 3347 samples, 2016-08-02 – 2016-08-04: 5054 samples, 2016-08-02 – 2016-08-05: 6602 samples, 2016-08-02 – 2016-08-06: 8786 samples.

## F  IMPACT OF SPATIAL AND TEMPORAL RESOLUTIONS

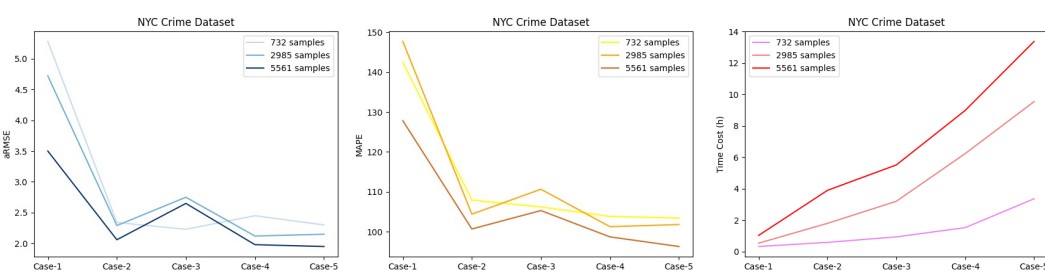

Figure 11: Impact of spatial and temporal resolution on model performance and computational efficiency. We vary the resolution for 5 cases: Case-1: 2 (12 hour) time grids, 25 (5 * 5) region blocks, Case-2: 4 (6 hour) time grids, 100 (10 * 10) region blocks, Case-3: 6 (4 hour) time grids, 225 (15 * 15) region blocks, Case-4: 8 (3 hour) time grids, 400 (20 * 20) region blocks, Case-5: 12 (2 hour) time grids, 625 (25 * 25) region blocks. And we also vary the sample size for 732, 2985, and 5561 samples.

We have explored the impact of spatial and temporal grid resolutions on both model performance and computational costs. We take NYC Crime dataset for an example and the results are presented in Fig.11. With finer resolutions for the time grid and region blocks, the model is expected to capture event patterns more accurately and with greater granularity in time and location. However, our experimental results indicate that increasing the fine-grained spatial and temporal resolution does not significantly enhance the model performance. For instance, when comparing Case-5 with Case-2 using a dataset of 2985 samples, despite Case-5 having finer resolution, the aRMSE only decreases from 2.29 to 2.15, and the MAPE decreases from 104.43 to 101.83. This could be attributed to the overly detailed partitioning of time and space, leading to insufficient instances of events at each time-location pair, thereby impacting the model's effectiveness. Further validation of this observation is evident when varying the sample size within the same case. For Case-2, increasing the sample size from 732 to 5561 results in a more significant improvement in model performance, with the aRMSE decreasing from 2.34 to 2.06 and the MAPE decreasing from 107.94 to 100.72. This underscores the substantial impact of increasing dataset size on model effectiveness. Hence, the results presented in our paper reflect a trade-off in selecting resolution based on balancing model performance and the level of detail in capturing time-location pair patterns. In the revised version of our paper, we will incorporate these experiments to demonstrate the generality of our approach.

## G  THE IMPACT OF FEATURES ON THE HUMAN DECISION-MAKING PROCESS

In our experiment, we consider the severity of the crime, suspect race, and suspect gender as key categorical features. The utility score for each time-location pair is determined through regression analysis involving these features, with the coefficients reflecting the magnitude of impact on the utility score. For each expert, the utility score patterns are different, reflecting different preference patterns. We take NYC Crime dataset with original 732 samples (temporal and spatial resolutions are 4 time slots and $10 \times 10$ area blocks) as an example and the results are shown in Fig.12, Fig.13, and Fig.14.

To illustrate the "human decision process" in a more clear way, we take an example for the top-5 crime event time location-pair for different races of expert-1 with largest utility score. Distinct patterns emerge based on the time and location of crime events across various racial groups. Black suspects tend to engage in criminal activities during the early morning or late night hours, while White Hispanic suspects are less active in criminal activities during the early morning hours. The timing of criminal activities among suspects of other races is more varied. At the regional level, the concentration areas for criminal activities among suspects of different races vary significantly. By incorporating social norms or individual information of suspects into the utility function, our approach better captures the role of individual differences and human decision-making processes in engaging in criminal activities.

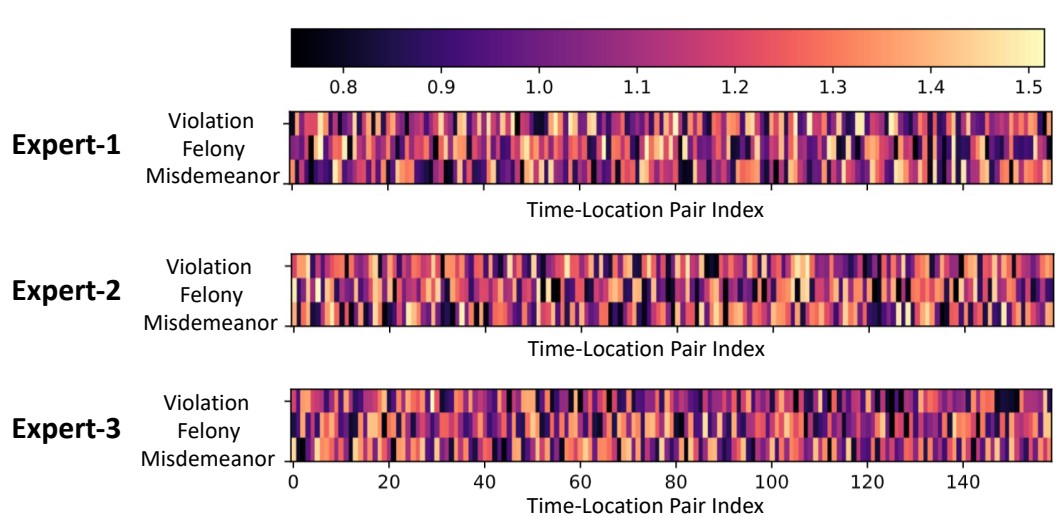

Figure 12: Impact of severity of the crime on utility function.

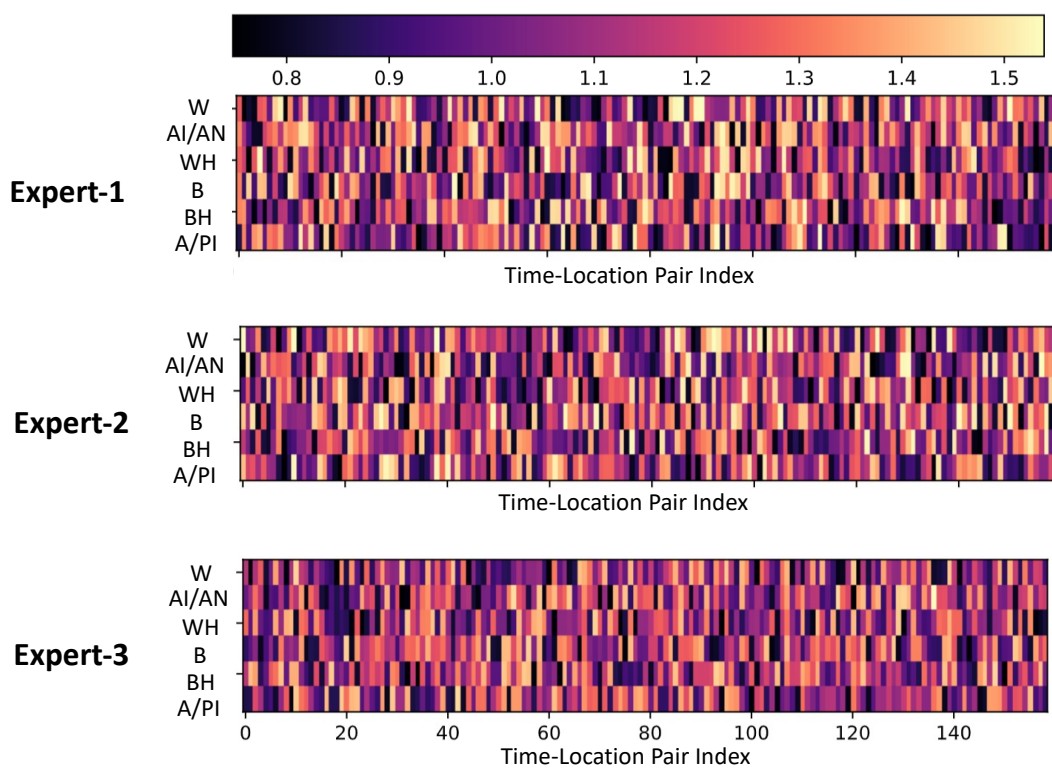

Figure 13: Impact of suspect race on utility function.

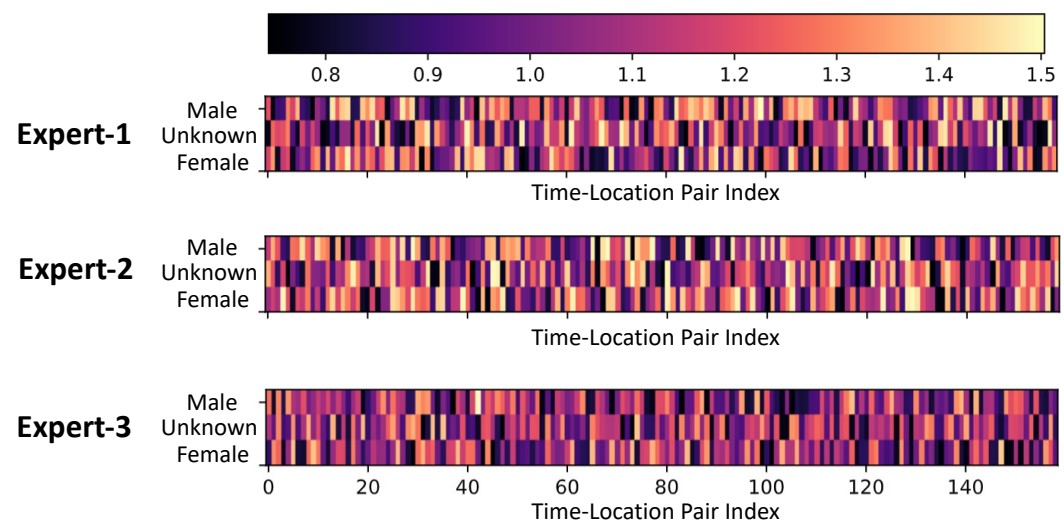

Figure 14: Impact of suspect gender on utility function.

| Race | W | AI/AN | WH |
|---|---|---|---|
| Top-1 | 0h-6h, (40.577, -73.845) | 12h-18h, (40.808, -73.895) | 12h-18h, (40.770, -73.794) |
| Top-2 | 18h-24h, (40.616, -74.045) | 0h-6h, (40.693, -73.995) | 18h-24h,(40.847, -73.845) |
| Top-3 | 0h-6h, (40.693, -73.845) | 0h-6h, (40.693, -73.995) | 6h-12h,(40.693, -73.945) |
| Top-4 | 12h-18h, (40.654, -73.945) | 18h-24h, (40.770, -73.895) | 6h-12h,(40.847, -73.845) |
| Top-5 | 18h-24h, (40.654, -73.895) | 12h-18h, (40.885, -73.895) | 12h-18h,(40.847, -73.845) |

Table 3: Top-5 crime event time location-pair for different races of expert-1 with largest utility score on NYC Crime Dataset. We use abbreviations to denote different races, where W: White, AI/AN: American Indian/Alaskan Native, and WH: White Hispanic. The time-location pairs are recorded as: time, (Lat., Lon.)

| Race | B | BH | A/PI |
|---|---|---|---|
| Top-1 | 0h-6h, (40.731, -73.995) | 18h-24h, (40.693, -73.744) | 0h-6h, (40.538, -74.196) |
| Top-2 | 0h-6h, (40.731, -73.895) | 0h-6h, (40.616, -74.146) | 0h-6h, (40.808, -73.845) |
| Top-3 | 0h-6h, (40.847, -73.895) | 0h-6h, (40.770, -73.995) | 18h-24h, (40.616, -73.795) |
| Top-4 | 18h-24h, (40.731, -73.995) | 6h-12h, (40.808, -73.845) | 12h-18h,(40.577, -73.995) |
| Top-5 | 0h-6h, (40.808, -73.845) | 18h-24h, (40.770, -73.945) | 12h-18h, (40.731, -73.945) |

Table 4: Top-5 crime event time location-pair for different races of expert-1 with largest utility score on NYC Crime Dataset. We use abbreviations to denote different races, where B: Black, BH: Black Hispanic, and A/PI: Asian/Pacific Islander. The time-location pairs are recorded as: time, (Lat., Lon.)

