# OpenReview forum: "Preference-Driven Spatial-Temporal Counting Process Models"
_ICLR.cc/2025/Conference — Submitted to ICLR 2025_

### Official Review · Reviewer_swpn · 2024-10-30

**Soundness:** 2
**Presentation:** 2
**Contribution:** 2
**Rating:** 3
**Confidence:** 4

**Summary:**

This paper aims at including human decision processes and social influence to observe criminal event counts. The proposed model is ambitious to include multiple human decision-making aspects, but the details of formulation and examination are missing. The experimental setup needs further reference to show its practicality.

**Strengths:**

S1. The experiments are conducted with three real-datasets.
S2. The writing is fluent and easy to understand.

**Weaknesses:**

W1. Overall, the major concerns are that the paper may not be self-contained and appears disconnected. First, although Fig. 1 visualizes the structure of the proposed model, most details explaining each part are not presented. For instance, what are the differences between spatial and position info? Which model does each expert use? Second, although the abstract and introduction state that social norms, environmental cues, and various other factors are considered, there is no corresponding formulation in Section 3. Finally, the experimental results do not validate these claims either. It is suggested to connect the claims with detailed descriptions in the methods and experiment sections.

W2. Please include up-to-date related works in top journals [1][2][3]. Moreover, half of the comparative baselines in the experiment section were published more than 10 years ago, which may be too outdated for fair comparisons. It is suggested to compare with newer methods instead.
[1] Weichao Liang, Zhiang Wu, Zhe Li, Yong Ge: CrimeTensor: Fine-Scale Crime Prediction via Tensor Learning with Spatiotemporal Consistency. ACM Trans. Intell. Syst. Technol. 13(2): 33:1-33:24 (2022)
[2] Shuai Zhao, Ruiqiang Liu, Bo Cheng, Daxing Zhao: Classification-Labeled Continuousization and Multi-Domain Spatio-Temporal Fusion for Fine-Grained Urban Crime Prediction. IEEE Trans. Knowl. Data Eng. 35(7): 6725-6738 (2023)
[3] Weichao Liang, Jie Cao, Lei Chen, Youquan Wang, Jia Wu, Amin Beheshti, Jiangnan Tang: Crime Prediction With Missing Data Via Spatiotemporal Regularized Tensor Decomposition. IEEE Trans. Big Data 9(5): 1392-1407 (2023)

W3. The definitions of matrices A and B on line 227, page 5, and the purpose of formulating them are unclear. Specifically, what do the two matrices embed, respectively? Additionally, right before introducing these matrices, the model already includes positional, spatial, temporal, and feature embeddings. An alternative approach might be to directly feed these four embeddings to the experts, rather than combining them with the two matrices to avoid additional computational overhead. This raises questions about the necessity, purpose, and benefit of the intermediate matrix decomposition-based embedding method compared to a straightforward alternative.

W4. Please clarify the “ranking” concept in the gating function, starting from line 251 on page 5. Equations 7, 8, and the loss function at line 274 resemble a cross-entropy formulation, which is a classification-based metric rather than a ranking one. Additionally, I am uncertain whether ranking is appropriate in this scenario. Specifically, while predicting the time and place of a crime, a top-1 ranking for occurrence may not directly indicate that a crime is happening, as the probability could still be low. Therefore, relying on ranking rather than probability prediction may lead to false alarms and overreactions.

W5. The practicality of the experimental setup is questionable. In the New York Crime and Chicago Crime datasets, each city is divided into 100 areas, and daytime is segmented into 4 time slots. However, it is unclear how large each area is after division. Is there evidence or a reference supporting that the 100-block granularity is beneficial for real-world law enforcement? Similarly, dividing daytime into four 6-hour slots may not be sufficiently granular. Is there a reference justifying this setup? Furthermore, it would be interesting to see the model’s performance at finer granularities, with smaller areas and shorter time slots.

W6. The experimental results may not fully examine the authors' claims. While modeling the “human decision process” is a key focus, it is unclear how this is tested in the experiments. Are there specific sequential criminal events in the datasets? If so, does the proposed method successfully retrieve these sequences? How does the model demonstrate that its improvements are due to modeling the human decision process? Otherwise, if each crime is independent, how are the datasets suitable for examining causal relationships? In this context, could simple statistics identify criminal hotspots at specific time slots to yield similar results to those in Fig. 2? It is recommended to elaborate further on human decision modeling in the experiments.

**Questions:**

Please refer to W3 to W6.

---

> ### Author Response · Authors · 2024-11-23
> **Response to Reviewer swpn**
>
> We first thank reviewer swpn for the insightful comments, especially for the questions about the details of our proposed model, which helped us to clarify our paper further. We would like to address the concerns one by one.
>
> In response to your mentioned weaknesses:
>
> **W1**: We summarize our response into following aspects:
>
> - **Explain more about spatial and positional embeddings**: We treat spatial and position information for different embeddings as the work of [1]. In our approach, spatial information, akin to patch information in [1] which breaks down the image (area region in our problem) into smaller patches (blocks in our problem), treating them as individual tokens similar to words in text, allowing the embedding to capture spatial information. The position information provides information about the locations of patches (blocks) within the region, helping the model understand the relative positions of different patches (blocks) and enabling it to learn spatial dependencies across the image (area region).
>
> - **Which model does each expert use?**: As indicated in Equation 5, we employ the same model architecture but utilize different parameters for each expert.
>
> - **Key individual features**:  Social norms and other pertinent factors are encapsulated within the utility function, formulated in a regression form. We have provided experiment fundings and detailed analysis in our response to **W6**. You can found our reply below.
>
> [1] Dosovitskiy, A. (2020). An image is worth 16x16 words: Transformers for image recognition at scale. arXiv preprint arXiv:2010.11929.
>
> **W2**: : Thanks for your suggestion about analyzing more related works. We will ensure to incorporate the reference you listed into the related work section of our paper. However, the codebases for the referenced methodologies have not been released. To ensure a fair comparison with state-of-the-art models, we have incorporated three more SOTA baseline models.
>
> - **HintNet** [2]: It performs a multi-level spatial partitioning to separate sub-regions with different risks and learns a deep network model for each level using spatio-temporal and graph convolutions.
> - **STNSCM** [3]: a causality-based interpretation model for the bike flow prediction.
> - **UniST** [4]:  a universal model designed for general urban spatio-temporal prediction across a wide range of scenarios.
>
> | Dataset | NYC Crime |  | Chicago Crime |  | Shanghai Mobike |  |
> |---|---|---|---|---|---|---|
> | Metric | aRMSE | MAPE | aRMSE | MAPE | aRMSE | MAPE |
> | HintNet [2] | 2.53 +/- 0.28 | 110.45 +/- 4.67 | 3.28 +/- 0.16 | 108.48 +/- 3.58 | 3.50 +/- 0.15 | 168.23 +/- 8.83 |
> | STNSCM [3] | 2.58 +/- 0.20 | 112.33 +/- 5.25 | 2.93 +/- 0.33 | 98.57 +/- 4.37 | 3.12 +/- 0.26 | 152.38 +/- 9.23 |
> | UniST [4] | 2.26 +/- 0.12 | 105.23 +/- 5.32 | 2.84 +/- 0.13 | 96.23 +/- 5.21 | 3.35 +/- 0.33 | 162.29 +/- 8.43 |
> | Ours* | 2.34 +/- 0.05 | 107.94 +/- 4.27 | 2.19 +/- 0.04 | 87.38 +/- 3.12 | 3.28 +/- 0.04 | 154.62 +/- 7.05 |
>
> Compared to the newly added baselines, our model surpasses all baselines in prediction tasks on the Chicago dataset. In the NYC dataset, our model demonstrates competitive performance with UniST and outperforms the other two models. The mean aRMSE and mean MAPE across three random runs of our model on the NYC dataset are 2.34 and 107.94, slightly higher than UniST's 2.26 and 105.23, respectively. However, our model exhibits a lower standard deviation, indicating better stability in our predictions. For the Mobike dataset, our model remains the second-best performing model, with a mean aRMSE and mean MAPE of 3.28 and 154.62, respectively, trailing only behind STNSCM with 3.12 and 152.38. Additionally, our model also exhibits lower standard deviation than the STNSCM model. Overall, our model significantly outperforms the models considered in our paper. Against the newly introduced state-of-the-art baselines, our model also achieves stable predictions and competitive results across all three datasets.
>
> [2] An, B., Vahedian, A., Zhou, X., Street, W. N., & Li, Y. (2022). Hintnet: Hierarchical knowledge transfer networks for traffic accident forecasting on heterogeneous spatio-temporal data. In Proceedings of the 2022 SIAM International Conference on Data Mining (SDM) (pp. 334-342). Society for Industrial and Applied Mathematics. \
> [3] Deng, P., Zhao, Y., Liu, J., Jia, X., & Wang, M. (2023, June). Spatio-temporal neural structural causal models for bike flow prediction. In Proceedings of the AAAI conference on artificial intelligence (Vol. 37, No. 4, pp. 4242-4249). \
> [4] Yuan, Y., Ding, J., Feng, J., Jin, D., & Li, Y. (2024, August). Unist: a prompt-empowered universal model for urban spatio-temporal prediction. In Proceedings of the 30th ACM SIGKDD Conference on Knowledge Discovery and Data Mining (pp. 4095-4106).

---

> ### Author Response · Authors · 2024-11-23
> **Response to Reviewer swpn**
>
> **W3**: The matrices A and B encode the learnable embedded time-location information for all pairs across the various latent classes. The benefit of the intermediate matrix decomposition-based embedding is to capture the interaction impacts of different temporal spatial grids by utilizing the cross product of A and B. This approach enhances flexibility compared to merely combining A and B.
>
> **W4**: Our sparse gating function generates a sparse vector, which consists of weights for each temporal-spatial grid. This vector retains only the top-k most significant choices with nonzero weights. The ranking of these weights reflects the preference order among choices. The probability for each temporal spatial grid depends on both the ranking and the utility. The ranking mechanism is used to reduce the dimension, i.e., from a high dimension choice set to a much smaller k dimension choice set. Since we aim to use our modeling framework to effectively represent the human decision-making process. The ranking mechanism delineates the initial step in the decision-making process — selecting a subset of candidates. The second step involves evaluating the utility of each choice. Although each event is treated independently, it emerges from underlying mixtures exhibiting diverse causal patterns.
>
> **W5**: The method we applied in the paper for dividing time and regions is common practice, and the granularity of our time and region divisions is generally similar to methods used in other published works. ST-HSL [5] applies a 3 km by 3 km spatial grid unit to New York City and Chicago, resulting in the generation of 256 and 168 disjoint spatial regions, respectively, slightly more than our 100 disjoint spatial regions. SpatialRank [6] partitions the Chicago area into 500 m by 500 m square cells, while HintNet [7] divides the whole state of Iowa into 5 km by 5 km grids (the area closely aligns with our partitioning) and designates a single day as the appropriate time interval. Following the same approach, we partitioned the time horizon and regions in a similar manner.
>
> We would like to highlight that, according to the official Mobike dataset documentation, the peak hours on workdays are specified as 16:00-20:00. To better capture the peak hour pattern, we designated this time frame as a distinct time slot, resulting in the division of the entire day into 6 time slots, each spanning 4 hours.
>
> Moreover, we have also added more experiments to investigate the impact of spatial and temporal grid resolutions on both model performance and computational costs. Please refer to our responses for **Reviewer v1uj** and **Reviewer VmVn, W3 & Q3**. Detailed experiment results and corresponding analysis can also be found in **Appendix F** in our revised paper.
>
> [5] Li, Z., Huang, C., Xia, L., Xu, Y., & Pei, J. (2022, May). Spatial-temporal hypergraph self-supervised learning for crime prediction. In 2022 IEEE 38th international conference on data engineering (ICDE) (pp. 2984-2996). IEEE. \
> [6] An, B., Zhou, X., Zhong, Y., & Yang, T. (2024). SpatialRank: urban event ranking with NDCG optimization on spatiotemporal data. Advances in Neural Information Processing Systems, 36. \
> [7] An, B., Vahedian, A., Zhou, X., Street, W. N., & Li, Y. (2022). Hintnet: Hierarchical knowledge transfer networks for traffic accident forecasting on heterogeneous spatio-temporal data. In Proceedings of the 2022 SIAM International Conference on Data Mining (SDM) (pp. 334-342). Society for Industrial and Applied Mathematics.

---

> ### Author Response · Authors · 2024-11-23
> **Response to Reviewer swpn**
>
> **W6**: We summarize our response into following aspects:
> - **Are there specific sequential criminal events in the datasets? If so, does the proposed method successfully retrieve these sequences?**: Specific sequential criminal events, such as repeated offenses by the same suspect, are rare in our crime dataset. However, the Mobike dataset documents multiple instances of bike rentals by the same user. The table below illustrates the mixture patterns adjusted by utility scores for different experts corresponding to 4 times bike rental records by the same user (user id 10344) on August 7, 2016, in Shanghai. The results indicate that under the same expert, the mixture patterns are similar, whereas they vary significantly across different experts. The mixture patterns generated by expert-1 exhibit obviously higher values compared to those produced by other experts. This demonstrates our model's ability to capture diverse user behaviors, underlying thinking processes, and sequential decision-making patterns in bike rental scenarios.
> | Time-Location Pair (time, (Lat., Lon.)) | Expert-1 | Expert-2 | Expert-3 | Expert-4 |
> |---|---|---|---|---|
> | 12h-16h, (31.185, 121.468) | 2.68*1e-2 | 1.48*1e-2 | 1.65*1e-2 | 1.63*1e-2 |
> | 16h-20h, (31.185, 121.468) | 3.25*1e-2 | 1.83*1e-2 | 1.92*1e-2 | 1.88*1e-2 |
> | 16h-20h, (31.185, 121.468) | 3.25*1e-2 | 1.83*1e-2 | 1.92*1e-2 | 1.88*1e-2 |
> | 20h-24h, (31.185, 121.468) | 2.40*1e-2 | 1.76*1e-2 | 1.96*1e-2 | 1.80*1e-2 |
>
> - **If each crime is independent, how are the datasets suitable for examining causal relationships?**: We assume there are K latent mixtures, and each crime belongs to one of the mixtures independently. The causal relationships are revealed by the latent mixtures.
>
> - **How does the model demonstrate that its improvements are due to modeling the human decision process? Could simple statistics identify criminal hotspots at specific time slots to yield similar results to those in Fig. 2?**: For Figure 2, simple statistics consider each time interval independently. However, our model jointly considers all the time intervals within a day and the interaction effects between time and location. To test the difference between the model performance of our model and statistical methods, we have added extra experiments. For statistical methods, we use the frequency of the previous day to predict the next day’s events. Seeing from the results shown in the table below, our model outperforms the statistical method on prediction tasks, indicating the incorporation of human decision processes of our proposed model indeed improves model performance.
>
> | Dataset | NYC Crime |  | Chicago Crime |  | Shanghai Mobike |  |
> |---|---|---|---|---|---|---|
> | Metric | aRMSE | MAPE | aRMSE | MAPE | aRMSE | MAPE |
> | Statistical Method | 4.75 +/- 0.00 | 128.93 +/- 0.00 | 7.34 +/- 0.00 | 132.54 +/- 0.00 | 7.82 +/- 0.00 | 175.54 +/- 0.00 |
> | Ours* | 2.34 +/- 0.05 | 107.94 +/- 4.27 | 2.19 +/- 0.04 | 87.38 +/- 3.12 | 3.28 +/- 0.04 | 154.62 +/- 7.05 |

---

> ### Author Response · Authors · 2024-11-23
> **Response to Reviewer swpn**
>
> **Continuing with W6**:
> - **For our claim of human decision process**: In our experiment, we consider the severity of the crime, suspect race, and suspect gender as key categorical features. The utility score for each time-location pair is determined through regression analysis involving these features, with the coefficients reflecting the magnitude of impact on the utility score. For each expert, the utility score patterns are different, reflecting different preference patterns. We have taken the NYC Crime dataset as an example and the results are shown in heatmaps in Appendix G of our revised paper.
>
> To illustrate the "human decision process" in a more clear way, we take an example for the top-5 crime event time location-pair for different races of expert-1 with largest utility score. The results are reported in the table below and the time-location pairs are recorded in "_time, (Lat., Lon.)_" format. From the results one can see that distinct patterns emerge based on the time and location of crime events across various racial groups. Black suspects tend to engage in criminal activities during the early morning or late night hours, while White Hispanic suspects are less active in criminal activities during the early morning hours. The timing of criminal activities among suspects of other races is more varied. At the regional level, the concentration areas for criminal activities among suspects of different races vary significantly. By incorporating social norms or individual information of suspects into the utility function, our approach better captures the role of individual differences and human decision-making processes in engaging in criminal activities.
>
> | Race | W | AI/AN | WH | B | BH | A/PI |
> |---|---|---|---|---|---|---|
> | Top-1 | 0h-6h, (40.577, -73.845) | 12h-18h, (40.808, -73.895) | 12h-18h, (40.770, -73.794) | 0h-6h, (40.731, -73.995) | 18h-24h, (40.693, -73.744) | 0h-6h, (40.538, -74.196) |
> | Top-2 | 18h-24h, (40.616, -74.045) | 0h-6h, (40.693, -73.995) | 18h-24h, (40.847, -73.845) | 0h-6h, (40.731, -73.895) | 0h-6h, (40.616, -74.146) | 0h-6h, (40.808, -73.845) |
> | Top-3 | 0h-6h, (40.693, -73.845) | 0h-6h, (40.693, -73.995) | 6h-12h, (40.693, -73.945) | 0h-6h, (40.847, -73.895) | 0h-6h, (40.770, -73.995) | 18h-24h, (40.616, -73.795) |
> | Top-4 | 12h-18h, (40.654, -73.945) | 18h-24h, (40.770, -73.895) | 6h-12h, (40.847, -73.845) | 18h-24h, (40.731, -73.995) | 6h-12h, (40.808, -73.845) | 12h-18h, (40.577, -73.995) |
> | Top-5 | 18h-24h, (40.654, -73.895) | 12h-18h, (40.885, -73.895) | 12h-18h, (40.847, -73.845) | 0h-6h, (40.808, -73.845) | 18h-24h,(40.770, -73.945) | 12h-18h, (40.731, -73.945) |

---

> > ### Comment · Reviewer_swpn · 2024-11-26
> > **Thank you for the responses**
> >
> > Thank you for providing the clarifications. To enhance the paper's reproducibility and clarity, it would be helpful to incorporate the above discussions into the manuscript. Additionally, the justifications for matrices A and B could be further strengthened. It might be beneficial to include additional discussions or experiments to better illustrate the necessity and impact of their flexibility.

---

### Official Review · Reviewer_cFeC · 2024-11-01

**Soundness:** 3
**Presentation:** 3
**Contribution:** 3
**Rating:** 6
**Confidence:** 4

**Summary:**

This paper presents a novel framework that integrates choice theory with social intelligence to model spatial-temporal counting processes, such as crime occurrences and bike-sharing activities. By capturing latent human preference patterns through utility functions, the model aims to provide deeper insights into the mechanisms driving these events. Empirical evaluations using crime and bike-sharing datasets show that the proposed model offers high predictive accuracy and interpretability compared to existing methods, though potential limitations and future research directions are not extensively discussed.

**Strengths:**

1. **Innovative Approach**: The paper introduces an innovative framework that integrates choice theory with social intelligence to model spatial-temporal counting processes. This approach addresses the complex decision-making processes and social factors influencing human-generated event data, such as crime occurrences and bike-sharing activities.
2. **Interpretable Insights**: The model provides interpretable insights by uncovering latent human preference patterns through utility functions. This feature helps in understanding the underlying mechanisms driving the observed event counts, which is valuable for both academic and practical purposes.
3. **Predictive Performance**: Empirical evaluations using crime and bike-sharing datasets show that the proposed model achieves good predictive accuracy compared to existing methods. The results indicate that the model can effectively predict event patterns and offer useful insights.
4. **Theoretical Foundation**: The paper derives a generalization bound that is independent of the number of latent classes, providing a theoretical foundation for the model's robustness and reliability. This theoretical contribution adds to the academic value of the work.
5. **Practical Flexibility**: The model demonstrates flexibility in handling different types of spatial-temporal data and can incorporate external interventions, making it adaptable to various real-world scenarios.

**Weaknesses:**

1. **Interpretability Validation**: While the model emphasizes interpretability, this claim is not fully supported with detailed case studies or qualitative analyses. More concrete examples and validation are needed to ensure that the insights provided are actionable and meaningful. Without such validation, the interpretability aspect, though highlighted as a strength, remains somewhat abstract and less convincing.
2. **Computational Efficiency**: The paper does not extensively address the computational efficiency of the model. Practical applications often involve large-scale datasets, and understanding the model's scalability and resource requirements is crucial. Without this information, it is challenging to determine the feasibility of deploying the model in real-world settings, which could limit its practical utility.
3. **Future Research Directions**:
The paper does not clearly outline future research directions or potential extensions of the model. Discussing these aspects would provide a clearer path for advancing the field and addressing current limitations. Identifying open questions and suggesting avenues for further investigation would enhance the paper's contribution and encourage ongoing research in this area.

**Questions:**

1. How do hyperparameter changes, such as learning rate, regularization parameters, and the number of mixture components, affect the model's performance?
2. In what ways can the model be tested on a variety of datasets with different spatial and temporal characteristics to assess its generalizability?
3. How can cross-validation and out-of-sample testing be conducted to ensure the model's stability and consistency?

---

> ### Author Response · Authors · 2024-11-23
> **Response to Reviewer cFeC**
>
> We appreciate that reviewer cFeC has a positive impression of our work. To address your concerns about the our method, we provide point-wise responses as follows.
>
> In response to your mentioned weaknesses:
>
> **W1**: Take crime datasets for examples, we consider the suspect features such as severity of the crime event, suspect race, and suspect gender as key categorical features to build the utility functions. And the utility score for each time-location pair is determined through regression analysis involving these features, with the coefficients reflecting the magnitude of impact on the utility score. For each expert, the utility score patterns are different, reflecting different preference patterns. By doing this, our paper's claim regarding the interpretability of the human decision process can be effectively demonstrated, as distinct suspect features reveal varying crime preferences across different time-location pairs. In our response to **Reviewer swpn, W6**, we present the experimental results. Please refer to our response for a comprehensive view of the results and corresponding analysis.
>
> **W2 & Q2**:  Please refer to our responses for **Reviewer v1uj**. We have also incorporated detailed experiment results and corresponding analysis in **Appendix E** and **Appendix F** of our revised paper.
>
> **W3**: Future research could integrate attention mechanisms into the gating function of choice model. This integration may enhance the model’s flexibility, enabling it to capture a broader range of and long-term information through neural networks. To improve the interpretability, we can also consider integrating the attention mechanisms into the utility function. These ideas serve as promising starting points which will be the future research direction to improve our work.

---

> ### Author Response · Authors · 2024-11-23
> **Response to Reviewer cFeC**
>
> In response to your questions:
>
> **Q1**: We use training negative log-likelihood, test aRMSE, test MAPE, and training time cost as **metric** to select appropriate hyper-parameters. Taking NYC-crime dataset with 732 samples as an example. And we divide the time horizon and region into 4 time slots and 100 region blocks.
>
> - **Impact of Learning Rate**
> | Learning Rate | 1e-4 | 5e-4 | 1e-3 | 5e-3 | 1e-2 | 5e-2 |
> |---|---|---|---|---|---|---|
> | Mean Neg. Log-likelihood | -4.980 | -5.112 | -5.059 | -4.839 | -4.854 | -5.010 |
> | aRMSE | 2.38 +/- 0.04 | 2.25 +/- 0.08 | 2.34 +/- 0.05 | 2.53 +/- 0.26 | 2.57 +/- 0.64 | 2.48 +/- 0.40 |
> | MAPE | 108.33 +/- 5.12 | 106.90 +/- 4.67 | 107.94 +/- 4.27 | 115.47 +/- 7.32 | 113.50 +/- 7.83 | 110.33 +/- 6.12 |
> | Time Cost (h) | 0.8354 | 0.7239 | 0.5859 | 0.5630 | 0.6239 | 0.5253 |
>
> We vary the learning rate through grid search from 1e-4 to 5e-2 and our current choice is 1e-3. When using a relatively low learning rate like 1e-4, the mean converged negative log-likelihood (lower is better) and the predictive accuracy remains nearly the same as our current selection (1e-3), but with increased computational cost for convergence. With a larger learning rate, the model performance starts to become unstable. For instance, when the learning rate is set to 5e-2, the mean aRMSE and mean MAPE for the prediction task across three random runs increase to 2.48 and 110.33, respectively, with larger standard deviations of 0.40 and 6.12. And the mean converged negative log-likelihood is also larger than the result achieved by our current choice of learning rate. Meanwhile, the training time for the model to converge does not decrease significantly. Therefore, we opted for a learning rate of 1e-3 in our experiments.
>
> - **Impact of Number of Experts**
> | Number of Experts | 1 | 2 | 3 | 4 | 5 |
> |---|---|---|---|---|---|
> | Mean Neg. Log-likelihood | -4.885 | -4.934 | -5.059 | -5.219 | -5.320 |
> | aRMSE | 2.52 +/- 0.08 | 2.46 +/- 0.10 | 2.34 +/- 0.05 | 2.25 +/- 0.12 | 2.28 +/- 0.08 |
> | MAPE | 113.85 +/- 5.23 | 110.54 +/- 5.67 | 107.94 +/- 4.27 | 105.83 +/- 4.50 | 104.73 +/- 4.75 |
> | Time Cost (h) | 0.4087 | 0.4860 | 0.5859 | 0.8216 | 1.2621 |
>
> For the number of experts, we vary it from 1 to 5. As our current choice for the NYC dataset is 3, decreasing it would degrade the model performance. However, increasing the number of experts does not yield a substantial improvement in model effectiveness but notably escalates computational costs. For instance, raising the number of experts from 3 to 5 would extend the training time from 0.5859 hours to 1.2621 hours because of increased learnable model parameters, while the negative log-likelihood, aRMSE, and MAPE metrics only marginally shift from -5.059 to -5.320, 2.34 to 2.28, and 107.94 to 104.73, respectively. Hence, to strike a balance between model performance and training efficiency, we opt to maintain the number of experts at 3 for the NYC dataset. We also employed a similar strategy when selecting the number of experts for other datasets.
>
> **Q3**: To ensure a model's stability and consistency, we can use k-fold cross-validation and out-of-sample testing. To implement K-Fold Cross-Validation, we need to split our crime or mobike datasets into k subsets and train the model on k-1 folds and validate on the remaining fold. We need to repeat this process k times, each time using a different fold as the validation set. After training the model using cross-validation like k-fold cross-validation, we can use out-of-sample testing to test it on entirely new, unseen data. In the prediction tasks detailed in our paper, we have employed both cross-validation and out-of-sample testing methodologies to validate and ensure the robustness of our model's performance.

---

### Official Review · Reviewer_VmVn · 2024-11-01

**Soundness:** 2
**Presentation:** 3
**Contribution:** 2
**Rating:** 3
**Confidence:** 5

**Summary:**

This paper is about the prediction problem for spatial-temporal event data generated by humans.  The authors introduced a framework integrating choice theory with social intelligence to model and analyze counting processes. The authors further conducted experiments on several real-world spatio-temporal datasets, and empirical evaluation of crime and bike-sharing datasets demonstrated that the proposed model could achieve the best performance.

**Strengths:**

1. The studied forecasting problem of spatio-temporal events is very important, interesting, and of high value in the real world.
2. The presentation is overall good, and the organization makes the paper easy to read and comprehend.
3. The authors select two representative metrics, aRMSE and MAPE, on which the proposed method achieves the best performance among all these models.

**Weaknesses:**

1. The datasets are small, leading to convincing results and conclusions. Although the authors have considered three datasets, NYC Crime, Chicago Crime, and Shanghai Mobike, for evaluation, the scales of these datasets are quite limited. There are only less than 1000 events on the first two datasets, which makes us wonder whether the proposed method can be used in real-world applications where the dataset may be very huge.
2. The technical contribution of the proposed method is questionable. The proposed method introduces a strategy of MoE, which is widely used in model ensembling and limits the contribution of the whole framework. In other words, it is very likely to improve performance by adding the MoE module. In short, the proposed solution is a bit straightforward.
3. Figure 2, Figure 3, and Figure 4 require improvement. Observing some informative and insightful conclusions from these figures is very hard since the grids are coarse-grained.

**Questions:**

Please answer the questions corresponding to the weaknesses mentioned above.
1. Why use these small datasets for evaluation? What about the actual value of the proposed method when applied to large-scale datasets?
2. How do you explain the performance improvement of the MoE module and the relation between it and the overall performance improvement?
3. How about the performance improvement when we have fine-grained spatial grids?

---

> ### Author Response · Authors · 2024-11-23
> **Response to Reviewer VmVn**
>
> We are grateful for your careful reading and useful suggestions! Below, we will address your concerns one by one.
>
> In response to your mentioned weaknesses and questions:
>
> **W1 & Q1**: Thanks for your insightful suggestion! We have added a scalability experiment to test the performance of our proposed model in large-scale datasets. Please refer to our response for **Reviewer v1uj** for the complete results across all three dataset we used in our paper. We have also incorporated the scalability experiments in **Appendix E** in our revised paper. Here we provided the corresponding analysis to address your concerns.
>
> Across all experiments, as the dataset sample size increases, both evaluation metrics, aRMSE and MAPE, generally decrease for the prediction task. For the NYC dataset, with an increase in data size from the current 732 samples to 5561 samples,  mean aRMSE over three random runs decreases from 2.34 to 2.06, and mean MAPE decreases from 107.94 to 100.72. For the Chicago dataset, with an increase in data size from the current 861 samples to 5321 samples, mean aRMSE over three random runs decreases from 2.19 to 1.93, and mean MAPE decreases from 87.38 to 85.50. For the Mobike dataset, to obtain datasets with varying sample sizes while ensuring consistency in the differences between sample sizes across datasets, we reselected data from August 2, 2016, to August 6, 2016. The results indicate that with an increase in data size from 1457 samples (one day) to 8786 samples (five days), the mean aRMSE over three random runs sharply decreases from 4.27 to 2.88, and the mean MAPE decreases from 172.50 to 148.67.
>
> The training time required for model convergence remains within acceptable ranges on the current computing infrastructure (details provided in Appendix D). For the large-scale NYC dataset with 5561 samples, the model converges in only 3.8970 hours. For the large-scale Chicago dataset with 5321 samples, the model converges in around 4.0410 hours. Even for the large-scale Mobike dataset with 8786 samples, our model converges and achieves good inference and prediction results after training for approximately 7.1056 hours, showcasing good scalability of our proposed model.
>
> **W2 & Q2**: Our approach aims to uncover the preference-driven decision-making processes. The mixture of experts is used to capture the thinking patterns of different latent groups. In addition to the Mixture of Experts module, our model incorporates choice theory and a sparse gating function to effectively capture the intricate patterns in spatial-temporal event data. Notably, for certain datasets, the mixture patterns may not be distinctly observable. Our model can capture these subtle differences and thus enhance the model performance. For instance, as illustrated in Figure 7 of the appendix, the first two mixtures from the Chicago dataset exhibit considerable similarity, while the third mixture contributes minimally with weight 0.1056. Guided by these complex mixtures, as demonstrated in Table 2, for the Chicago dataset, our model's performance significantly surpasses that of the baseline models.

---

> ### Author Response · Authors · 2024-11-23
> **Response to Reviewer VmVn**
>
> **W3 & Q3**: Your concern about the resolution is constructive! Employing a finer resolution undoubtedly enhances the model's persuasiveness by capturing more detailed time-location patterns for event occurrences. But we want to emphasize that the method we applied in the paper for dividing time and regions is common practice, and the granularity of our time and region divisions is generally similar to methods used in other published works. ST-HSL [1] applies a 3 km by 3 km spatial grid unit to New York City and Chicago, resulting in the generation of 256 and 168 disjoint spatial regions, respectively, slightly more than our 100 disjoint spatial regions (as we reported in our paper). SpatialRank [2] partitions the Chicago area into 500 m × 500 m square cells, while HintNet [3] divides the whole state of Iowa into 5 km by 5 km grids (the area closely aligns with our partitioning) and divides a single day as the appropriate time interval.
>
> Moreover, we have explored the **impact of spatial and temporal grid resolutions** on both model performance and computational costs. Please refer to our response for **Reviewer v1uj** for the complete experiment results. Detailed experiment results and corresponding analysis can also be found in **Appendix F** in our revised paper.
>
> With finer resolutions for the time grid and region blocks, the model is expected to capture event patterns more accurately and with greater granularity in time and location. However, our experimental results indicate that increasing the fine-grained spatial and temporal resolution does not significantly enhance the model performance. For instance, when comparing Case-4 with Case-2 using a dataset of 2985 samples, despite Case-4 having finer resolution, the mean aRMSE (over three different runs) only decreases from 2.29 to 2.12, and the mean MAPE decreases from 104.43 to 101.30. This could be attributed to the overly detailed partitioning of time and space, leading to insufficient instances of events at each time-location pair, thereby impacting the model's effectiveness. Further validation of this observation is evident when varying the sample size within the same case. For Case-2, increasing the sample size from 2985 to 5561 results in a more significant improvement in model performance, with the mean aRMSE decreasing from 2.29 to 2.06 and the mean MAPE decreasing from 104.43 to 100.72. It addresses the substantial impact of increasing dataset size on model effectiveness. Therefore, the results presented in our paper reflect a trade-off in selecting resolution based on balancing model performance and the level of detail in capturing time-location pair patterns.
>
> [1] Li, Z., Huang, C., Xia, L., Xu, Y., & Pei, J. (2022, May). Spatial-temporal hypergraph self-supervised learning for crime prediction. In 2022 IEEE 38th international conference on data engineering (ICDE) (pp. 2984-2996). IEEE. \
> [2] An, B., Zhou, X., Zhong, Y., & Yang, T. (2024). SpatialRank: urban event ranking with NDCG optimization on spatiotemporal data. Advances in Neural Information Processing Systems, 36. \
> [3] An, B., Vahedian, A., Zhou, X., Street, W. N., & Li, Y. (2022). Hintnet: Hierarchical knowledge transfer networks for traffic accident forecasting on heterogeneous spatio-temporal data. In Proceedings of the 2022 SIAM International Conference on Data Mining (SDM) (pp. 334-342). Society for Industrial and Applied Mathematics.

---

### Official Review · Reviewer_v1uj · 2024-11-04

**Soundness:** 3
**Presentation:** 3
**Contribution:** 2
**Rating:** 6
**Confidence:** 2

**Summary:**

This paper presents a new spatial-temporal counting process model that integrates choice theory and social intelligence to capture human decision-driven event occurrences, such as crime rates and bike-sharing usage. The core idea is to use latent utility functions to represent diverse decision-making factors and to apply a mixture-of-experts model with a sparse gating function for adaptive selection. The model aims to reveal underlying patterns in counting processes, providing both predictive power and interpretability.

**Strengths:**

The paper is methodologically sound, with a well-defined approach supported by both theoretical and empirical analyses. The experimental setup is robust, including multiple real-world datasets, and the model's performance is compared against established baselines to highlight its predictive strength.

The paper is well-structured and provides comprehensive explanations of its key components, including the latent utility functions, mixture-of-experts model, and gating function. Diagrams and formulas aid in clarifying complex concepts, making the model's framework accessible for readers.

This framework contributes significantly to spatial-temporal modeling, especially in domains where human decision-making drives event occurrences. By enabling a nuanced understanding of preference-driven behavior and offering predictive power, the model has applications in fields like criminology, urban planning, and shared mobility systems.

**Weaknesses:**

The use of mixture-of-experts and the sparse selection mechanism may raise concerns regarding computational scalability when applied to large-scale, high-dimensional spatial-temporal data. While the model performs well on mid-sized datasets, it is unclear if the sparse gating function and multiple experts could handle significantly larger spatial grids or finer temporal resolutions without substantial computational costs. A discussion on computational efficiency or optimization strategies, such as parallelization, would strengthen the model’s applicability to broader scenarios.

**Questions:**

Please refer to weaknesses.

---

> ### Author Response · Authors · 2024-11-23
> **Response to Reviewer v1uj**
>
> We thank the reviewer for your careful reading and insightful comments! In response to the reviewers' suggestion, we have expanded our evaluation to include multiple datasets with varying sample sizes to test the **scalability of our proposed model**. Detailed experiment results and corresponding analysis can be found in **Appendix E** in the revised paper.
>
> - **Scalability experiments on NYC Crime Dataset**
> | Sample Size | 732 | 1840 | 2985 | 4016 | 5561 |
> |---|---|---|---|---|---|
> | aRMSE | 2.34 +/- 0.05 | 2.25 +/- 0.06 | 2.29 +/- 0.13 | 2.10 +/- 0.08 | 2.06 +/- 0.10 |
> | MAPE | 107.94 +/- 4.83 | 103.32 +/- 4.67 | 104.43 +/- 5.33 | 101.83 +/- 4.33 | 100.72 +/- 5.67 |
> | Time Cost (h) | 0.5859 | 1.3252 | 1.7971 | 2.5254 | 3.8970 |
>
> - **Scalability experiments on Chicago Crime Dataset**
> | Sample Size | 861 | 2434 | 3207 | 4578 | 5321 |
> |---|---|---|---|---|---|
> | aRMSE | 2.19 +/- 0.04 | 2.56 +/- 0.08 | 2.12 +/- 0.12 | 2.25 +/- 0.08 | 1.93 +/- 0.10 |
> | MAPE | 87.38 +/- 3.12 | 94.67 +/- 3.83 | 86.23 +/- 5.22 | 90.94 +/- 4.95 | 85.50 +/- 4.12 |
> | Time Cost (h) | 0.7785 | 1.6995 | 2.2787 | 3.2730 | 4.0410 |
>
> - **Scalability experiments on Shanghai Mobike Dataset**
> | Sample Size | 1457 | 3347 | 5054 | 6602 | 8786 |
> |---|---|---|---|---|---|
> | aRMSE | 4.27 +/- 0.12 | 3.57 +/- 0.08 | 3.06 +/- 0.12 | 2.85 +/- 0.12 | 2.88 +/- 0.67 |
> | MAPE | 172.50 +/- 8.48 | 158.33 +/- 8.25 | 151.54 +/- 7.33 | 148.34 +/- 7.50 | 148.67 +/- 8.19 |
> | Time Cost (h) | 1.4245 | 2.4924 | 3.8226 | 5.5305 | 7.1056 |
>
> For NYC dataset, the date time was divided into four time slots, and the New York area was segmented into 100 small area blocks
> based on longitude and latitude. We use the same temporal and spatial resolution for Chicago dataset. For Shanghai Mobike, we divide the date time into 6 time grids and divide the Shanghai area into 100 area blocks. Across all experiments, as the sample size increases, both evaluation metrics, aRMSE and MAPE, decrease for the prediction task. Taking the NYC dataset as an example, with an increase in data size from the current 732 samples to 5561 samples,  mean aRMSE (over three different seeds) decreases to 2.06, and MAPE decreases to 100.72. The training time required for model convergence remains within acceptable limits on the current computing infrastructure (details provided in Appendix D). For the NYC dataset with 5561 samples, the model converges in only 3.8970 hours. Even for the large-scale Mobike dataset with 8786 samples, our model converges and achieves good inference and prediction results after training for approximately 7.1056 hours. For all these datasets, the model's prediction accuracy increases with larger sample size, requiring more training time, yet within reasonable range.

---

> ### Author Response · Authors · 2024-11-23
> **Response to Reviewer v1uj**
>
> Additionally, we have explored the **impact of spatial and temporal grid resolutions** on both model performance and computational costs. The results are presented in the tables below and detailed experiment results and corresponding analysis can be found in **Appendix F** in the revised paper. We use NYC crime dataset with 2985 samples and 5561 samples across 4 cases, each case with different time grid length and resolution for region. The descriptions about these 4 cases and corresponding experiment results are as below:
>
> - **Case-1**: 2 time grids (12 hour per grid), 25 region blocks (5 * 5)
> - **Case-2**: 4 time grids (6 hour per grid), 100 region blocks (10 * 10)
> - **Case-3**: 6 time grids (4 hour per grid), 225 region blocks (15 * 15)
> - **Case-4**: 8 time grids (3 hour per grid), 400 region blocks (20 * 20)
>
> | Case | Case-1 |   | Case-2 |   | Case-3 |   | Case-4 |   |
> |---|---|---|---|---|---|---|---|---|
> | Sample Size | 2985 | 5561 | 2985 | 5561 | 2985 | 5561 | 2985 | 5561 |
> | aRMSE | 4.72 +/- 0.25 | 3.50 +/- 0.12 | 2.29 +/- 0.13 | 2.06 +/- 0.10 | 2.75 +/- 0.06 | 2.65 +/- 0.08 | 2.12 +/- 0.17 | 1.98 +/- 0.15 |
> | MAPE | 147.67 +/- 8.33 | 127.83 +/- 6.25 | 104.43 +/- 5.33 | 100.72 +/- 5.67 | 110.67 +/- 2.65 | 105.33 +/- 3.57 | 101.30 +/- 3.76 | 98.74 +/- 5.12 |
> | Time Cost (h) | 0.5346 | 1.0295 | 1.7971 | 3.8970 | 3.2025 | 5.5152 | 6.2380 | 8.9856 |
>
> With finer resolutions for the time grid and region blocks, the model is expected to capture event patterns more accurately and with greater granularity in time and location. However, our experimental results indicate that increasing the fine-grained spatial and temporal resolution does not significantly enhance the model performance. For instance, when comparing Case-4 with Case-2 using a dataset of 2985 samples, despite Case-4 having finer resolution, the mean aRMSE (over three different seed) only decreases from 2.29 to 2.12, and the mean MAPE decreases from 104.43 to 101.30. This could be attributed to the overly detailed partitioning of time and space, leading to insufficient instances of events at each time-location pair, thereby impacting the model's effectiveness. Further validation of this observation is evident when varying the sample size within the same case. For Case-2, increasing the sample size from 2985 to 5561 results in a more significant improvement in model performance, with the aRMSE decreasing from 2.29 to 2.06 and the MAPE decreasing from 104.43 to 100.72. This underscores the substantial impact of increasing dataset size on model effectiveness. Hence, the results presented in our paper reflect a trade-off in selecting resolution based on balancing model performance and the level of detail in capturing time-location pair patterns.

---

### Author Response · Authors · 2024-12-03

Dear Reviewers, Senior Area Chairs, Area Chairs, and Program Chairs,

We are deeply grateful for the insightful comments and suggestions, which are invaluable for enhancing our work. We are excited that the reviewers hold positive feedback and find our work “The paper is methodologically sound, well-structured and provides comprehensive explanations of its key components, and contributes significantly to spatial-temporal modeling.” (Reviewer v1uj), “The studies forecasting problem of spatial-temporal events is very important, interesting, and of high value in the real world.” (Reviewer VmVn), “This paper provides an innovative approach with interpretable insights, good predictive performance, theoretical foundation, and practical flexibility” (Reviewer cFeC), and “The experiments are thorough and the writing is fluent and easy to understand” (Reviewer swpn).

In our response, we have included additional clarifications and experiments. To ensure transparency and clarity, we have outlined our main responses as follows:

- **Scalability and computational cost**: We have expanded our evaluation to include multiple datasets with varying sample sizes to test the scalability of our proposed model. Detailed experiment results and corresponding analysis can be found in **Appendix E** in the revised paper. Encouragingly, for all these datasets, the model's prediction accuracy increases with larger sample size, requiring more training time, yet within reasonable range.

- **Impact of spatial and temporal resolution**: We have explored the impact of spatial and temporal grid resolutions on both model performance and computational costs. In **Appendix F** of the revised paper, we have provided the detailed experiment results and corresponding analysis. The results presented in our paper reflect a trade-off in selecting resolution based on balancing model performance and the level of detail in capturing time-location pair patterns.

- **Hyper-parameter selection**: We have used metrics such as training negative log-likelihood, training time, and prediction accuracy to select hyper-parameters such as learning rate and and number of experts. Experimental results can be found in our response to Reviewer cFeC.

- **Elaboration on human decision modeling**: In our experiments, the utility score for each time-location pair is determined through regression analysis involving key human features (such as gender and race), with the coefficients reflecting the magnitude of impact on the utility score. For each expert, the utility score patterns are different, reflecting different preference patterns. We have taken the NYC Crime dataset as an example and the results are shown in heatmaps in **Appendix G** of our revised paper. By incorporating social norms or individual information of human into the utility function, our approach better captures the role of individual differences and human decision-making processes in engaging in human activities.

- **Comparison with extra baselines**: As suggested by Reviewer swpn, we compare our model with three more SOTA baselines (HintNet[1], STNSCM[2], and UniST[3]). Compared with newly added SOTA baselines, our model achieves stable predictions and competitive results across all three datasets

In addition to the academic contributions, our method holds practical significance. Our proposed approach can capture how individual choices influence the distribution of events over time and space, helping identify overlooked tendencies in human activities. By recognizing preference patterns among different populations, it enables tailored planning and plays a crucial role in guiding human decision-making processes like crime control. We believe this innovative approach has the potential to inspire future research endeavors.

[1] An, B., Vahedian, A., Zhou, X., Street, W. N., & Li, Y. (2022). Hintnet: Hierarchical knowledge transfer networks for traffic accident forecasting on heterogeneous spatio-temporal data. In Proceedings of the 2022 SIAM International Conference on Data Mining (SDM) (pp. 334-342). Society for Industrial and Applied Mathematics. \
[2] Deng, P., Zhao, Y., Liu, J., Jia, X., & Wang, M. (2023, June). Spatio-temporal neural structural causal models for bike flow prediction. In Proceedings of the AAAI conference on artificial intelligence (Vol. 37, No. 4, pp. 4242-4249). \
[3] Yuan, Y., Ding, J., Feng, J., Jin, D., & Li, Y. (2024, August). Unist: a prompt-empowered universal model for urban spatio-temporal prediction. In Proceedings of the 30th ACM SIGKDD Conference on Knowledge Discovery and Data Mining (pp. 4095-4106).

---

### Meta-Review · Area_Chair_D4F7 · 2024-12-16

**Metareview:**

The paper introduces a framework to predict spatial-temporal event data generated by humans. In terms of strengths, the reviewers pointed out that the problem tackled by the paper is important, the proposed framework is methodologically sound, and the paper is well-written. In terms of weaknesses, the reviewers had concerns regarding the scalability of the framework, the claims regarding interpretability, the comparison with the state of the art, the significance of the technical contribution, and the experimental setup used in the experiments. Overall, two of the reviewers were fairly negative and two were mildly positive and, despite the authors made a significant effort in addressing the reviewers' concerns during the rebuttal period, the reviewers were not persuaded to change their overall recommendation. As a consequence, I cannot recommend to accept the paper.

**Additional Comments On Reviewer Discussion:**

The reviewers raised a number of concerns, summarized in the metareview. The authors made a significant effort in addressing these concerns, conducting additional experiments during the rebuttal period. One of the reviewers did follow up, however, this reviewer did not change their overall recommendation. The other reviewers did not follow up nor change their overall recommendation. As a consequence, I cannot recommend to accept the paper.

---

### Decision · Program_Chairs · 2025-01-22

Reject